# Transposable elements are regulated by context-specific patterns of chromatin marks in mouse embryonic stem cells

Jiangping He[1,2,3,4], Xiuling Fu[5], Meng Zhang[1,2,3,4,6], Fangfang He[5], Wenjuan Li[2,6], Mazid Md. Abdul[1,2,3,4,6], Jianguo Zhou[1,2,3,6], Li Sun[5], Chen Chang[5], Yuhao Li[5], He Liu[1], Kaixin Wu[1], Isaac A. Babarinde[5], Qiang Zhuang[5,7], Yuin-Han Loh[8,9], Jiekai Chen [1,2,3], Miguel A. Esteban [1,2,3,6] & Andrew P. Hutchins [5]

The majority of mammalian genomes are devoted to transposable elements (TEs). Whilst TEs are increasingly recognized for their important biological functions, they are a potential danger to genomic stability and are carefully regulated by the epigenetic system. However, the full complexity of this regulatory system is not understood. Here, using mouse embryonic stem cells, we show that TEs are suppressed by heterochromatic marks like H3K9me3, and are also labelled by all major types of chromatin modification in complex patterns, including bivalent activatory and repressive marks. We identified 29 epigenetic modifiers that significantly deregulated at least one type of TE. The loss of *Setdb1*, *Ncor2*, *Rnf2*, *Kat5*, *Prmt5*, *Uhrf1*, and *Rrp8* caused widespread changes in TE expression and chromatin accessibility. These effects were context-specific, with different chromatin modifiers regulating the expression and chromatin accessibility of specific subsets of TEs. Our work reveals the complex patterns of epigenetic regulation of TEs.

[1] Key Laboratory of Regenerative Biology of the Chinese Academy of Sciences and Guangdong Provincial Key Laboratory of Stem Cell and Regenerative Medicine, Guangzhou Institutes of Biomedicine and Health, Chinese Academy of Sciences, 510530 Guangzhou, China. [2] Joint School of Life Sciences, Guangzhou Medical University and Guangzhou Institutes of Biomedicine and Health, Chinese Academy of Sciences, 511436 Guangzhou, China. [3] Guangzhou Regenerative Medicine and Health-Guangdong Laboratory (GRMH-GDL), 510530 Guangzhou, China. [4] University of Chinese Academy of Sciences, 100049 Beijing, China. [5] Department of Biology, Southern University of Science and Technology, 1088 Xueyuan Lu, 518055 Shenzhen, China. [6] Laboratory of RNA, Chromatin, and Human Disease, Guangzhou Institutes of Biomedicine and Health, Chinese Academy of Sciences, 510530 Guangzhou, China. [7] State Key Laboratory of Medicinal Chemical Biology and College of Life Sciences, Nankai University, 300071 Tianjin, China. [8] Epigenetics and Cell Fates Laboratory, A*STAR Institute of Molecular and Cell Biology, 61 Biopolis DriveProteos, Singapore 138673, Singapore. [9] Department of Biological Sciences, National University of Singapore, Singapore 117543, Singapore. Correspondence and requests for materials should be addressed to A.P.H. (email: andrewh@sustc.edu.cn)

The modification of histones is an elaborate system to regulate gene expression, and provides an epigenetic landscape for the cell-type-specific interpretation of the genome. Yet, the major class of genomic elements in the cellular genome are not genes, but transposable elements (TEs), including endogenous retroviruses (ERVs)[1]. TEs were originally thought of as genetic parasites with roles in human disease[2], but TEs are now understood to contribute to normal biological processes[3]. There are many examples of exapted TEs that have become host cell genes, such as the RAG enzymes which are essential for antibody and T cell receptor recombination, or the Syncytin genes which are involved in placental development[3,4]. TEs can be transcribed to produce RNA, and have contributed to the evolution of long non-coding RNAs, microRNAs, and circular RNAs[5]. In addition, TEs often mimic host cell functions by incorporating cis-regulatory elements[6], which can recruit transcription factors (TFs) to promote TE activity. This was initially observed for the TF repressor REST[7], but has since been observed for a wide-range of TFs[8–10]. TEs copy themselves in the genome and are co-opted to form new regulatory elements[1,11,12], and contribute to the rewiring of gene regulatory networks[6]. However, much of this data on exaptation of TEs is derived from genomic data and there is argument over how much is functional[6].

TEs/ERVs are silenced by a range of molecular mechanisms, including heterochromatin formation[13–15], mRNA editing[16], and DNA methylation[17]. DNA methylation is thought to be the dominant suppressive mechanism in somatic tissues[17]. However, DNA is globally demethylated during early embryonic development, and TEs are released from repression in a controlled, stage-specific manner[18,19]. The TEs are then free to compete with the epigenetic suppression mechanisms to duplicate themselves and enter the germ line[20]. Consequently, there is a delicate balance between the beneficial effects of TEs, and their deleterious effects on genome integrity[6,21,22].

TEs are suppressed in embryonic cells in a process that is well described for ERVs. Zinc-finger proteins (ZFPs) bind to specific sequences in ERVs[23], recruit the adaptor protein TRIM28/KAP1, and the histone H3K9me3 methyltransferase SETDB1 to silence TEs[13,24–28]. In addition to H3K9me3, there are other modes of epigenetic suppression of TEs[29], such as the methylation of H4K20me3[30], H3K27me3[31], and H4R3me2[20], the biotinylation and sumoylation of H2A, H3, and H4 histones[32,33], and the deposition of the histone variant H3.3[34]. It is clear that the epigenetic system is regulating TEs[4,29,35–38], however, there are at least 1100 distinct types of TE, comprising millions of genomic copies, for which the epigenetic regulation is unclear.

Here, we reveal that TEs are marked by chromatin modifications in complex patterns. Of the 32 chromatin marks we explored, 22 were enriched on at least one TE type. We find evidence not only of repressive marks, but widespread marking of TEs by activatory marks, including bivalent marking of TEs by repressive H3K9me3, and activatory H3K27ac chromatin marks. When we knocked down a panel of chromatin regulators, 29 of 41 led to the deregulation of expression of at least one type of TE. The loss of *Setdb1*, *Ncor2*, *Rnf2*, *Kat5*, *Prmt5*, *Uhrf1*, and *Rrp8* led to widespread changes in the expression of TEs, and a corresponding change in chromatin accessibility. Finally, we explore the consequences of these observations by showing that the loss of chromatin modifiers induces a gene expression program similar to totipotent 2 cell (2C) embryos, characterized by the upregulation of MERVL TEs and 2C-specific genes. Overall, our data suggests that the chromatin modifying system manages the expression of TEs by context-specific deposition of chromatin marks that regulate the chromatin state and expression level of TEs.

## Results

**TEs are marked by overlapping patterns of chromatin marks**. To explore the chromatin code for the suppression of TEs, we uniformly reanalyzed mouse ESC ChIP/epigenetic-seq data for 32 chromatin marks, including histone methylation, acetylation, ubiquitination, and variant histones, along with DNA 5-methylcytosine (5mC), 5-hydroxymethylcytosine (5hmC), and Assay for Transposase-Accessible Chromatin (ATAC)-seq (Supplementary Data 1). Data was quality controlled by removing poorly correlating biological replicates (See Methods and Supplementary Figure 1a). A concern in the analysis of repeats is the use of multimapped reads. In our analysis, for multimapped reads only the highest scoring alignment was kept, and was randomly assigned to a TE copy. To reduce bias due to random distribution of reads across TE copies we considered TEs as metagenes that describe the entire set of TE copies across the genome. Below, we show some example genomic views, but it should be kept in mind that the analysis of individual TE copies can be ambiguous.

The ChIP-seq data comes from serum + LIF grown ESCs, which have relatively high levels of DNA methylation, particularly at Intracisternal A-particle (IAP) TEs[38–40]. However, many TEs have demethylated DNA, and ESCs use DNA methylation-independent mechanisms to suppress TEs. Knockout of all three DNA methyltransferases led to upregulation of mostly IAP-family TEs only[40], and this upregulation is rapidly compensated for by small RNA and polycomb-mediated repression[38,40]. Indeed, only 65/679 TE types had >2-fold enrichment of 5mC DNA methylation, and most of the methylated TEs were IAP or IAP LTRs (Long terminal-repeats), and a mixture of other ERVs, for example, BGLII_Mus (Fig. 1a, b, Supplementary Figure 1b and Supplementary Data 2).

The second major mechanism for the regulation of TEs in ESCs is the formation of heterochromatin, mediated by H3K9me3 and H4K20me3[14,41,42]. We found that 87/679 TE types were > 2-fold-enriched with H3K9me3, and 72/679 were marked by H4K20me3. H3K9me3 and H4K20me3 tended to co-occur, and they both marked 58/679 TE types (Supplementary Data 2). Examples are shown for the ERVK IAPLTR4_I, and IAPEY-int which were enriched for H3K9me3 and H4K20me3 (Fig. 1a, b).

Of the remaining TEs, some had simple patterns of chromatin modifications, such as MERVL TEs that were enriched only for H3K56ac (Fig. 1b). In total 41/679 TE types were enriched with a single chromatin mark (Supplementary Data 2). The most common pattern was for multiple chromatin marks on the TEs, as 117/679 had 2 or more chromatin marks, and, at a relaxed enrichment threshold of >1.5 fold, 198/679 TE types were enriched with at least 2 chromatin marks (Supplementary Data 2). Examples include RLTRETN_Mm, which was enriched for 14 of the 32 chromatin marks we profiled (Fig. 1a, b). Interestingly, TEs were marked by chromatin patterns associated with other genomic features. Many TEs were enriched with chromatin marks indicative of enhancers, such as H3K4me1 which marked 23/679 TE types. For example, a 3.0 fold-enrichment of H3K4me1 was found on the ERVK RLTR9D2 (Supplementary Data 2). The promoter mark H3K4me3 was enriched at 14/679 TEs, including the LINE (long interspersed nuclear element) L1Md_Gf (7.2-fold-enriched), and the ERVK RLTRETN_Mm (2.9 fold-enriched) (Fig. 1a, b and Supplementary Data 2). ERVKs/ERV1s were enriched with activatory chromatin marks, including histone acetylations, such as H3K27ac (RLTR13G >6.2-fold-enriched) and H2B120ac (MMERGLN_LTR >4.6 fold-enriched) (Fig. 1a, c). In total 75/679 TE types were marked by one or more acetylated histone mark and 42/679 by H3K27ac. To gain a global view of TE chromatin, we clustered the patterns of chromatin marks on TEs

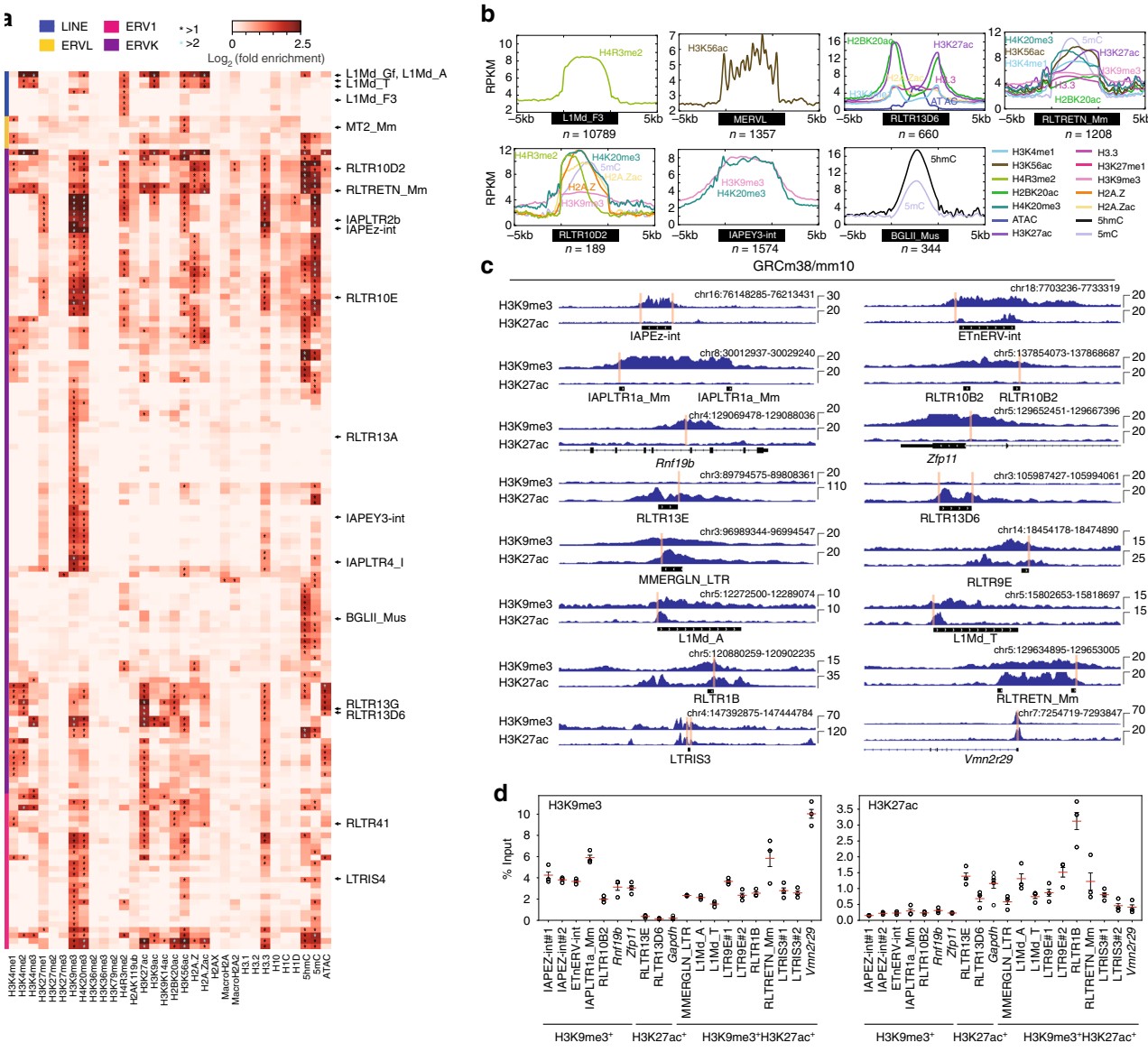

**Fig. 1** TEs are marked by TE-type-specific epigenetic signatures. **a** Heatmap of 32 chromatin marks in normal serum-grown ESCs and their fold-enrichment at the indicated TE types. The RPKM (reads per kilobase of TE per million library sequence reads) was measured for all copies of that TE type, and each TE type was treated as a metagene. Fold-enrichment over background was calculated by measuring the RPKM for each TE type, taking the maximum binned value, and comparing this to size matched, randomly selected genomic regions. A chromatin mark was considered enriched if it was > 2-fold enriched (marked with an asterisk). Those TE types > 4 fold-enriched are indicated with a cyan asterisk. See Supplementary Data 1 for detailed sample information. The full heatmap and table for all TE types is in Supplementary Figure 1b and Supplementary Data 2. **b** Read count tag density pileups (In RPKM) of the indicated chromatin marks for the selected TEs. The TEs were scaled to the same size, and the flanking 5 kb regions are shown. $n =$ indicates the number of TEs used in that pileup, for this and all subsequent pileup figures. **c**, **d** Native-ChIP-qPCR using antibodies against H3K27ac and H3K9me3 for selected TEs, measured as percent of input. The location of the primer pairs are indicated with red lines in the genomic views in (**c**). See Supplementary Data 7 for the primer pair sequences. Genomic locations are from the mm10 mouse assembly, for this and all other genomic figures. The scale of the profiles ranges from a baseline of 0 up to the indicated value, in normalized arbitrary units, for this and all subsequent genome views. Native-ChIP-qPCR (**d**) was performed three times in biological replicate each with three technical replicates, the open circles indicate the mean of the technical replicates, the red bar is the mean of the biological replicates, and the error bars are the standard error of the mean. Source data are provided as a Source Data file

(Supplementary Figure 1c). We could identify 6 major clusters of chromatin marks, including: a heterochromatic-cluster (H4K20me3, H3K9me3, H3.3), a H4Rme2-cluster (H4R3me2, 5mC, H1D, H1C), a transcriptional elongation-cluster (H3K36me3, H3K36me2), a variant/histone-polycomb-cluster (H3K27me3, H3K27me2, H3.1 and H3.2), a promoter-cluster (H3K4me3, H3K9ac), and an enhancer-like cluster (H3K4me1, H3K27ac, ATAC, H2BK20ac) (Supplementary

Figure 1c). Overall, of the 32 chromatin marks we analyzed 22 were enriched >2 fold on at least one TE type, and 27 were enriched >1.5 fold (Supplementary Data 2), indicating broad and complex patterns of chromatin marks on TEs.

LINE elements are a special case as they occupy 18% of the mouse genome[4], and are increasingly recognized for their important roles in embryogenesis and chromatin

organization[14,43,44]. The analysis of LINE TEs suggests that they are mainly enriched with H4R3me2 (Fig. 1a and Supplementary Data 3). However, LINEs can be divided into longer near-intact elements and shorter truncated versions. We split the LINE1 elements into long (>5 kb) and short (<5 kb), and measured the chromatin mark enrichment at the two classes of LINEs. Whilst both long and short LINEs were enriched for H4R3me2, the long LINEs were enriched for 15/32 chromatin marks (Supplementary Figure 1d). This included chromatin marks representing many genomic elements, including enhancers (H3K4me1, H3K27ac), promoters (H3K4me3), activatory regions (acetylated histones), DNA methylation (5mC and 5hmC), and heterochromatin (H3K9me3, H4K20me3). The chromatin marks tended to be located at the 5′ end of the LINEs (Supplementary Figure 1e). The chromatin at LINEs nonetheless remains difficult to analyze as LINEs occupy so much of the mouse genome. They are frequently close to other regulatory elements, and this can bias the chromatin patterns. Nonetheless, the pattern of chromatin marks on LINEs is complex, and highlights their roles in recruiting TFs[10], in chromatin organization[43], and cell type control[44].

We noticed that many TE types had both an activatory and a repressive chromatin mark. For example, 16/679 TE types had both H3K9me3 and H3K27ac (Supplementary Data 2). One explanation for the overlapping patterns of activatory and repressive chromatin marks is that individual TE copies have discreet chromatin marks. However, genome views and pileup heatmaps suggested that the chromatin marks were simultaneously marking the TE copies (Supplementary Figure 1f, g). These observations come with two important caveats. The first is that the ChIP-seq data was generated from pooled cells, and the marks could separately label different cells. The second caveat is related to uncertainty in mapping reads to specific TE copies due to the repetitive nature to TEs. To address the latter issue, we performed native-ChIP-qPCR for a selection of TEs, using primer pairs that amplified unique regions of the genome close to the TEs (Fig. 1c). As controls, we used primer pairs for IAPEZ-int, ETnERV-int, IAPLTR1a_Mm and RLTR10B2 (marked only by H3K9me3), and RLTR13E and RLTR13D6 (marked only by H3K27ac). In agreement with the genomic views, the TEs, L1Md_A/T, LTR9E, RLTR1B, RLTRETN_Mm, and LTRIS3, were indeed enriched for both H3K9me3 and H3K27ac, as measured by native-ChIP-qPCR (Fig. 1d). Overall, the native-ChIP-qPCR results closely matched the genome views (Fig. 1c), and indicated that, at least for this limited set of TEs, they are marked by both activatory and repressive chromatin marks.

**Chromatin modifiers are recruited to TEs**. TEs can function as regulatory elements, recruiting TFs and chromatin modifiers (CMs)[3,9,45]. To explore the role of CMs in depositing chromatin marks at TEs we collected 91 publically available ChIP-seq datasets for CMs in ESCs (Supplementary Data 1 and 3). In agreement with the patterns of chromatin marks at TEs, CMs were also bound to TEs in specific patterns (Fig. 2a). All 91 CMs were enriched >2 fold on at least 1 TE type, and 174/691 TE types were enriched with at least 1 CM, and of those, 134/174 were enriched with 2 or more CMs (Supplementary Data 3), for example RLTR46, which was bound by HCFC1, KANSL3, CHD8, and YY1 (Supplementary Figure 2a). TEs often contain DNA-binding motifs for TFs[8], which can recruit CMs. To explore the TF binding motifs inside TEs, we performed de novo motif discovery. TEs were rich in DNA-binding motifs (Supplementary Figure 2b), and most types of DNA-binding motif are represented. We noticed that the YY1 and REST motifs were enriched in our analysis. The YY1 motif was enriched at LINE, IAP, and

MERVL TEs, as previously reported[46], and ChIP-seq data for YY1 showed that YY1 was bound to IAPLTR2b (Supplementary Figure 2c). The REST motif was identified as enriched in RMER21A TEs, and ChIP-seq data indicated REST was binding to RMER21A (Supplementary Figure 2c). One motif that interested us was the NR5A2 motif, which was in almost all copies of RLTR13B2 (Supplementary Figure 2d). NR5A2 is interesting as it can reprogram primed EpiSCs to naive ESCs[47]. ChIP-seq data indicated that NR5A2 was bound to RLTR13B2 TEs (Supplementary Figure 2e), and RLTR13B2 was marked by a pattern of chromatin marks reminiscent of enhancers (H3K27ac, H3K4me1, and other activatory marks) (Supplementary Figure 2f). In addition to NR5A2 binding, there was a range of CMs bound, including mediator complex proteins, MED1, MED12, and NIPBL, and several co-activator/repressor complex proteins, including ASH2L, BRD4, HDAC1/2, KDM1A, P300, and SMARCA4 (Supplementary Figure 2g). Importantly, RNA-seq and ATAC-seq data indicated that Nr5a2 and RLTR13B2 were specifically expressed in naive ESCs, and that the RLTR13B2 TEs had open chromatin only in the naive ESCs (Supplementary Figure 2h, i). This analysis suggests that RLTR13B2 is under the epigenetic control of NR5A2, and may be acting as a cis-regulatory element for NR5A2. These results highlight the interactions between TEs, TFs, and CMs.

To interrogate the global arrangement of CMs, chromatin marks and TEs, we generated a relational network. Edges were drawn between the nodes (TEs/gray, chromatin marks/pink, or CM/green), if the chromatin mark or CM was enriched >2 fold on the TE type (Fig. 2b). One major group in the network was the IAP-family TEs which were marked by heterochromatic marks, such as H3K9me3, and H4K20me3[29,30] and the CMs SETDB1, SUV39H1/2, TRIM28, and ZFP809 (Fig. 2b–d). Many of the IAPs also had methylated DNA[48,49], and were bound by MBD-family proteins[50] (Fig. 2a). The chromatin structural protein CTCF was bound to RLTR46 TEs[9], along with its known protein partners RAD21, STAG1, and STAG2[51] (Fig. 2c). We also observed other CMs enriched at RLTR46 TEs, including HCFC1, KANSL3, CHD8, and YY1 (Supplementary Figure 2a; Supplementary Data 3). The widespread binding of CMs to a TE is similar to the pattern observed for LINE L1s in human cancer cell lines, which are bound by a large number of TFs/CMs, including CTCF[10]. Interestingly, we did not observe CTCF bound to L1, in agreement with a previous report[9], suggesting CTCF binding to L1 is species-specific.

Global co-correlation of the binding patterns of all CMs and chromatin marks on TEs revealed several clusters of co-occurring CMs/marks (Supplementary Figure 3). We annotated the clusters based on the function of known CM members. There were two supergroups which we designate the Heterochromatin group and the Active/Open group, supplemented by several smaller groups that were distinguished by known regulators/binders of TEs, such as DAXX[52], REST[7], or CTCF[9,10]. Amongst the Active/Open super-group were several subgroups which could be distinguished based upon their major members, such as the P300/mediator, or Pol II, along with groups that had patterns resembling enhancers (H3K4me1). One of the subgroups contained CHAF1A, a histone chaperone involved in depositing the variant histone H3.3, both of which have been implicated in TE control[24,34]. Finally, the Heterochromatin supergroup contained many factors specific for silencing, particularly SETDB1, MBD proteins and the chromatin marks H3K9me3 and H4K20me3. Overall, TEs were bound by a wide-range of CMs associated with both repressive and activatory chromatin (Fig. 2a, c, d and Supplementary Data 3), and these results demonstrate the interconnected binding patterns of the TEs by CMs through chromatin marks. However, although CMs bind promiscuously to TEs in a complex chromatin

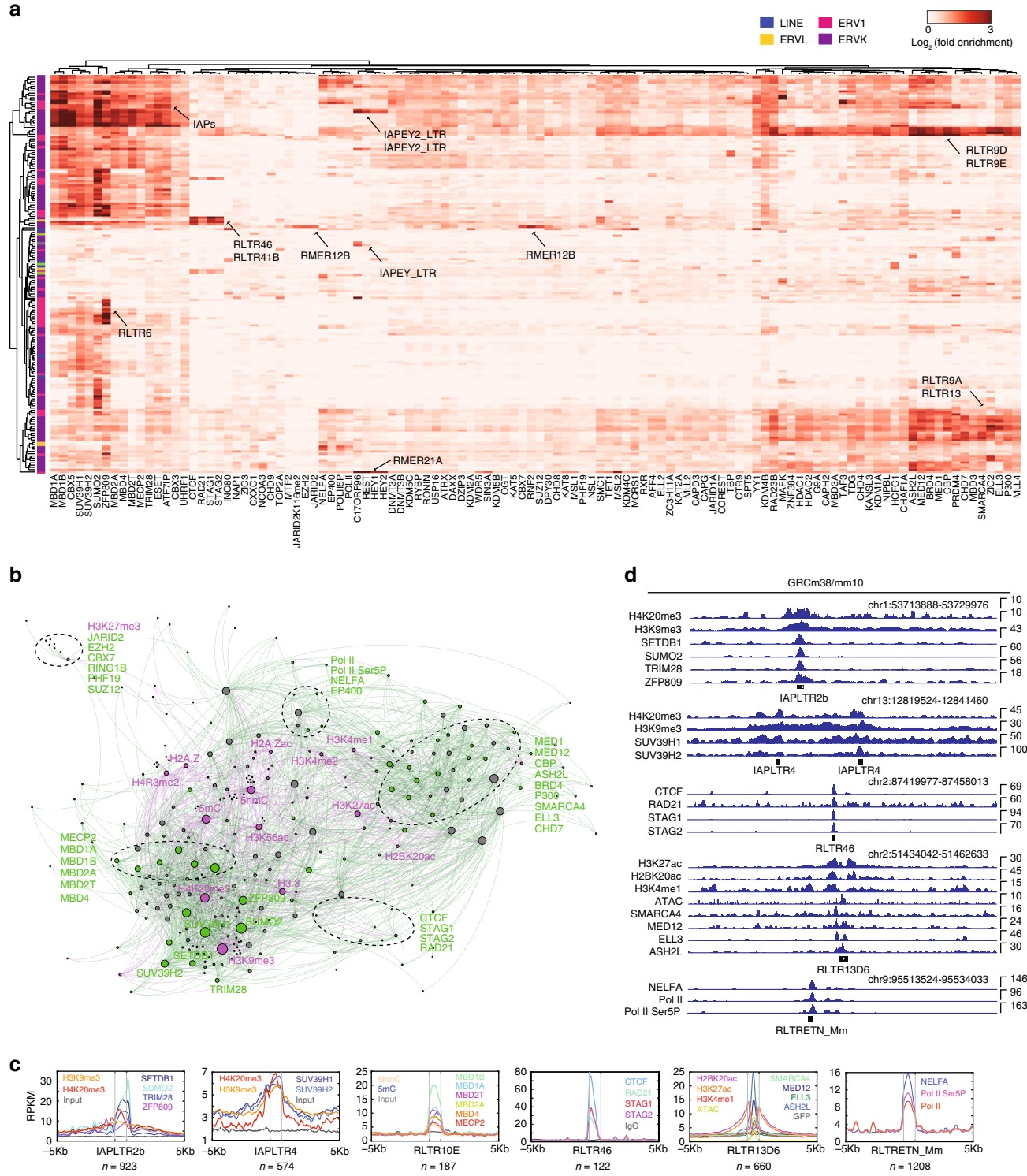

**Fig. 2** TEs are marked by type-specific patterns of chromatin modifiers. **a** The fold-enrichment of CMs bound to TEs was measured, as in Fig. 1a. The data were hierarchically clustered and selected TE types are indicated. The full table is in Supplementary Data 3. See Supplementary Data 1 for details of the samples used in this study. **b** Relational network for the chromatin marks (purple), the selected CMs (green), and TEs (gray). Edges were drawn between a chromatin mark and a TE, or a CM and a TE, if the chromatin mark or CM was enriched at least 2-fold. Node sizes are proportional to the network degree. Selected subgroups are indicated on the network. **c** Read count tag density pileups (In RPKM) of the indicated chromatin marks and CMs for the selected TEs. The TEs were scaled to the same size, and the flanking 5 kb regions are shown. The gray line indicates the input/IgG control. **d** Example genomic views for the indicated TEs, chromatin marks and CMs are indicated. Care should be taken in interpreting genomic views due to uncertainty in mapping reads to specific TE copies, this applies to all genome views in the manuscript

environment, it is unclear if they are directly regulating the expression and chromatin state of these TEs.

**Chromatin modifiers regulate type-specific TE expression**. To explore how CMs control TE expression, we selected a panel of reader, writer, and eraser CMs[53] (Supplementary Figure 4a) to knockdown, based on the clusters we detected in Fig. 2a, b and Supplementary Figure 3. As cells can compensate for the derepression of TEs by activating alternative suppressive mechanisms[38,40], we chose to knockdown over a short time period of four days. This decision has two benefits. First, it allowed us to study those CMs that are lethal if knocked down for a prolonged time. Second, it should minimize changes in cell fate which would complicate the results, as the cells reconfigure their chromatin in response to differentiation. In total, we knocked down 41 CMs, and subjected them to RNA-seq. To rule out a change in cell fate we measured the expression of a panel of pluripotency and differentiation marker genes. Their expression indicated that most knockdowns had little influence on cell fate (Supplementary Figure 4b, c and Supplementary Data 4). Principal component analysis gave similar results, as the knockdown samples did not diverge from the control samples (Supplementary Figure 4d). Analyzing the number of differentially expressed genes came to a similar conclusion, as minimal numbers of genes were differentiated in each knockdown (Supplementary Figure 4e). There was, however, two notable exceptions. Knockdown of *Mcrs1* showed an upregulation of chromatin-related genes and a loss of germ cell character, and *Chd4*, which showed signs of differentiation towards the immune system (Supplementary Figure 4f). We include *Chd4* and *Mcrs1* in the analysis below, but it should be noted that these two knockdowns are differentiating, and interpretation of their effect on TEs is complicated. For all other knockdowns, there was little evidence of differentiation, and no loss of pluripotency. This data set provides a system to look at the short-term deregulation of TEs when CMs are lost, in the absence of major cell type changes.

We next explored if these CM knockdowns can deregulate TE expression. We aligned the RNA-seq data to LINE, SINE (short interspersed nuclear element), LTR (ERV), and DNA class TEs (See Methods). On average 1–5% of reads would map to TEs (Supplementary Figure 5a), and we observed higher percentages of reads mapping to TEs in the knockdown samples (most >2% versus 1.6% in the sh*Luc* control), suggesting a higher level of TE-derived RNA (Supplementary Figure 5a). As expected, knockdown of *Setdb1* caused a robust upregulation of many types of TE (Fig. 3a, b). In addition, 29/41 other CM knockdowns significantly up or down-regulated at least 1 type of TE (Fig. 3a), and 8 knockdowns upregulated at least 1 type of TE >4 fold (Fig. 3b). We used RT-qPCR to validate a selection of TE expression changes in knockdowns with shRNAs against *Rnf2*, *Setdb1*, *Ncor2*, *Prmt5*, and *Ash2l* (Supplementary Figure 5b).

We selected eight CM knockdowns (and sh*Ncor1*, as a control) that deregulated 20 or more types of TE for further analysis. Western blotting for the pluripotency proteins OCT4 and NANOG confirmed that the eight selected knockdowns were not inducing differentiation (Supplementary Figure 5c). Knockdown of these eight CMs led to a variety of effects on the expression of TEs. Knockdown of *Setdb1* deregulated only a few TE types, but it led to the largest change in the magnitude TE expression. Conversely, knockdown of *Ncor2* or *Rnf2* deregulated more types of TE, but the magnitude of upregulation was reduced (Figs. 3a, b). There was surprisingly little overlap in the TE types that were upregulated, and most TE types were specific to a single knockdown (Supplementary Figure 5d, e).

TEs can span a wide-range of evolutionary time, and some TEs are active, or recently active, whilst others are inactive and ancient. Importantly, young and old TEs are regulated differently[12]. We measured the evolutionary age of TEs, and found that younger TEs tended to have higher enrichment of chromatin marks, although the correlation was modest (Supplementary Figure 1b). Younger TEs were more likely to be deregulated in a CM knockdown, although, again, the correlation was modest (Supplementary Figure 5f). These results suggest that younger TEs are more likely to be under active epigenetic regulation.

We next investigated the relationship between TE expression changes and chromatin marks. There was no global alteration in the levels of histone modifications, with the exception of the *Rnf2* knockdown, which reduced H2AK119ub and H3K27ac (Fig. 3c). This indicated that the changes in histone modifications were context-specific, as we saw previously with the overexpression of HDAC3 in ESCs which did not alter global H3K27ac levels, but did alter context-specific H3K27ac sites[54]. There was a direct relationship between CMs and chromatin mark. For example, knockdown of *Setdb1* led to significant upregulation of TEs high in H3K9me3, whilst knockdown of the DNA methylation cofactor *Uhrf1* led to upregulation of TEs high in 5mC, particularly IAPs (Fig. 3d, e, and Supplementary Figure 5g). Knockdown of *Prmt5* has previously been demonstrated to upregulate LINE L1 and IAP elements in primordial germ cells, and ground state 2i + LIF ESCs[20], and we observe the same in serum + LIF grown ESCs (Fig. 3f, g, h). The LINE and IAP TEs upregulated in the *Prmt5* knockdown were enriched with H4R3me2, the catalytic target for PRMT5, indicating a link between PRMT5 and H4R3me2 at TEs (Figs. 1a, 3g, and Supplementary Figure 1b, d, e). In addition to these associations, we also found new links between CMs and chromatin marks. Knockdown of *Hdac2* and *Hdac5* upregulated TEs enriched with H3K56ac, H3K9me3, and H4K20me3, and knockdown of *Ncor2* upregulated TEs enriched with H3K9me3 and H3K56ac (Supplementary Figure 5g, h). NCOR2 is a co-repressor[54], so the knockdown of *Ncor2* activating TEs may seem unusual, but it is not uncommon for co-repressors to have activatory function. For example, the co-repressor SIN3A, a binding-partner for NCOR2, can function as a co-activator in ESCs[55]. Overall, these observations indicate a complex code of CMs regulating sets of TEs by altering chromatin marks.

**Chromatin modifiers alter chromatin accessibility at TEs**. The loss of CMs leads to context-specific deregulation of TEs, which in many cases was associated with specific chromatin marks. However, whilst many CMs are specific epigenetic writers/erasers (e.g., SETDB1 writes H3K9me3, RNF2 writes H2AK119ub), they are also members of large multi-protein complexes involved in multiple regulations (e.g., RNF2 is part of polycomb repressor complex 1), or there is substantial redundancy (e.g., SETDB1 and SUV39H1 both target H3K9me3). Consequently, the number of potential chromatin mark changes caused by the loss of any single CM can be large. To gain an overview of changes in chromatin in response to the knockdown of a CM, we used ATAC-seq[56,57], to probe TE chromatin accessibility after knockdown. Of the 142,119 non-redundant peaks in the ATAC-seq data, we could detect changes in 15,076 loci (Supplementary Figure 6a, b). Chromatin that became accessible in each shRNA knockdown showed matching upregulation of genes within 10 kb of the ATAC-seq peak (Supplementary Figure 6c, d). The changes in chromatin accessibility were also reflecting chromatin marks. For example, ATAC peaks that became accessible when *Prmt5* was knocked down were enriched for H4R3me2. A similar pattern was observed for *Setdb1* knockdown and H3K9me3 (Supplementary Figure 6e). These results show that changes in accessibility are a suitable proxy for chromatin changes.

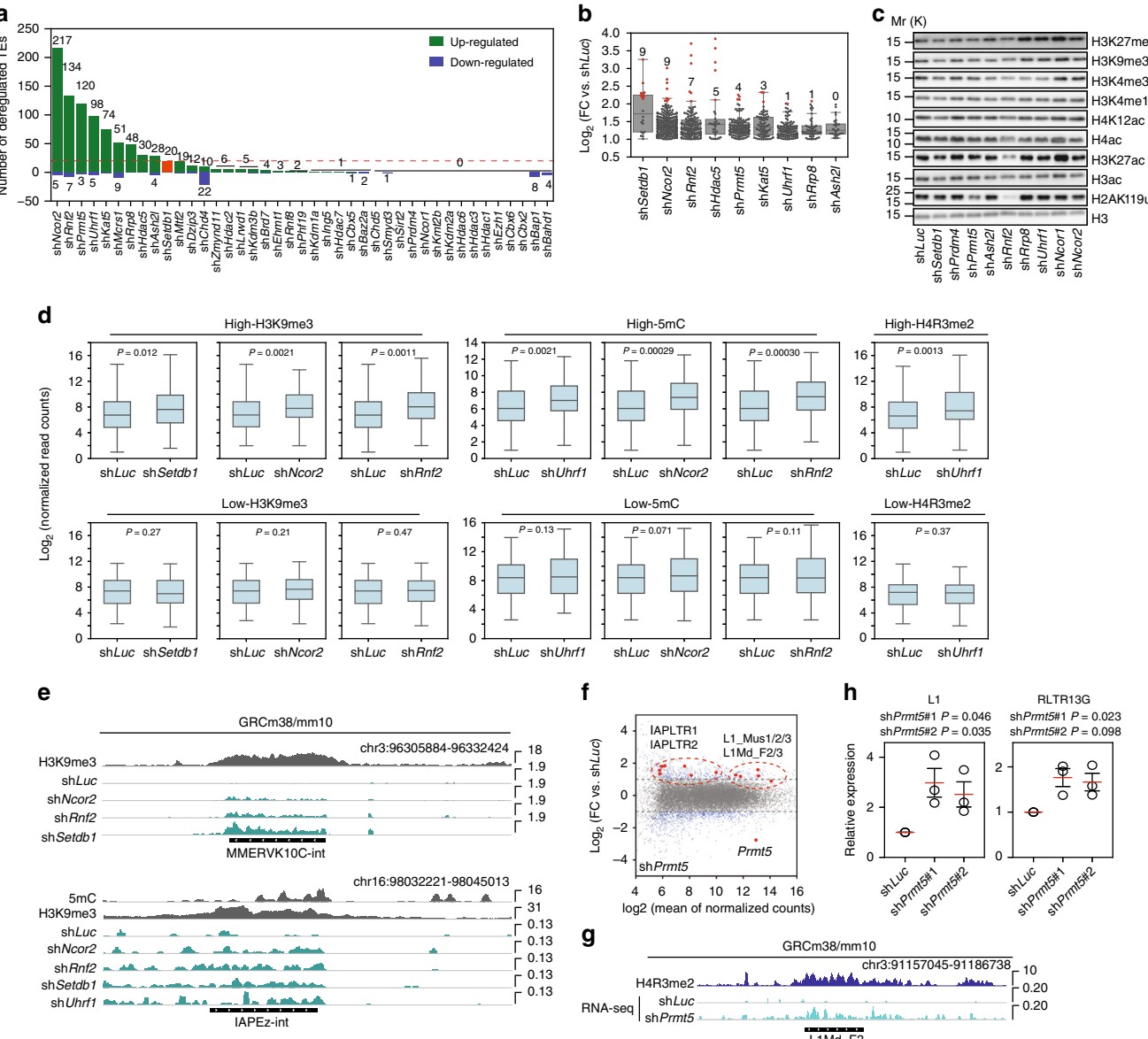

**Fig. 3** Chromatin modifiers regulate the type-specific expression of TEs. **a** Number of significantly deregulated TE types for the indicated shRNA knockdown (*P* values < 0.05, Benjamini & Hochberg corrected, absolute fold-change >2, relative to sh*Luc*). See Supplementary Data 1 for detailed sample information. **b** Boxplots of log2 (fold-change (FC) versus sh*Luc*) for all significantly deregulated TE types. In the boxplot, outliers represent the mean of a TE type. TE types significantly upregulated > 4 fold are indicated in red and the number is marked. **c** Western blot of chromatin marks, from the indicated shRNA knockdowns. Molecular weight is indicated as Mr (K). This experiment was performed twice with similar results. **d** Boxplots of the expression level for all TE types with either High or Low levels of the indicated chromatin mark. High was defined as those TE types with > 2-fold enrichment of the indicated chromatin mark. Low sites were selected by ranking the TEs by fold enrichment, and taking TEs from the bottom of the list until it was the same length as the High list. **e** Genomic views of MMERVK10C-int and IAPEz-int showing chromatin marks and RNA-seq in the indicated shRNA knockdown. **f** MA-scatter (log-fold-change, mean average expression) plot showing sh*Prmt5* knockdown relative to sh*Luc*. Blue dots are significantly different genes or TEs (*P* value < 0.05, Benjamini & Hochberg corrected, absolute fold-change > 2, relative to sh*Luc*). Red dots indicate IAP or LINE1 elements, or the *Prmt5* gene. RNA-seq data was from 2 biological replicates. **g** Genome view of H4R3me2 ChIP-seq data and sh*Prmt5* RNA-seq data for L1Md_F3 LINE1. **h** RT-qPCR expression of the LINE1 L1Md_F2, and RLTR13G, relative to sh*Luc* control knockdown. *P* values are from a one-tailed Student's *t*-test. The circles are the mean of three technical replicates, the red bar is the mean of the three biological replicates, and the error bars are the standard error of the mean. Source data are provided as a Source Data file. Primers used are described in Supplementary Data 7. For the boxplots, the midline indicates the median, boxes indicate the upper and lower quartiles and the whiskers indicate 1.5*interquartile range

We measured the chromatin accessibility at TEs. 153 TE types showed an increase in chromatin accessibility (Fig. 4a, Supplementary Data 5). These changes were mainly specific to a single knockdown, as 108 TE types were uniquely upregulated, whilst 42 TE types were upregulated by more than one shRNA knockdown (Fig. 4a, b, and Supplementary Data 5). Importantly, opening of chromatin at TEs, was matched by significant upregulation of the

corresponding RNA expression (Fig. 4c). This pattern extended to individual types of TE, for example, the chromatin opened and RNA was expressed at MT2_Mm, when *Rnf2* was knocked down (Fig. 4d). Similarly, at LINE1 L1Md_F2 TEs, chromatin opened and expression was upregulated when *Prmt5* was knocked down (Fig. 4d, e). This suggests a direct relationship between L1Md_F2, and regulation of expression by PRMT5 through modification of

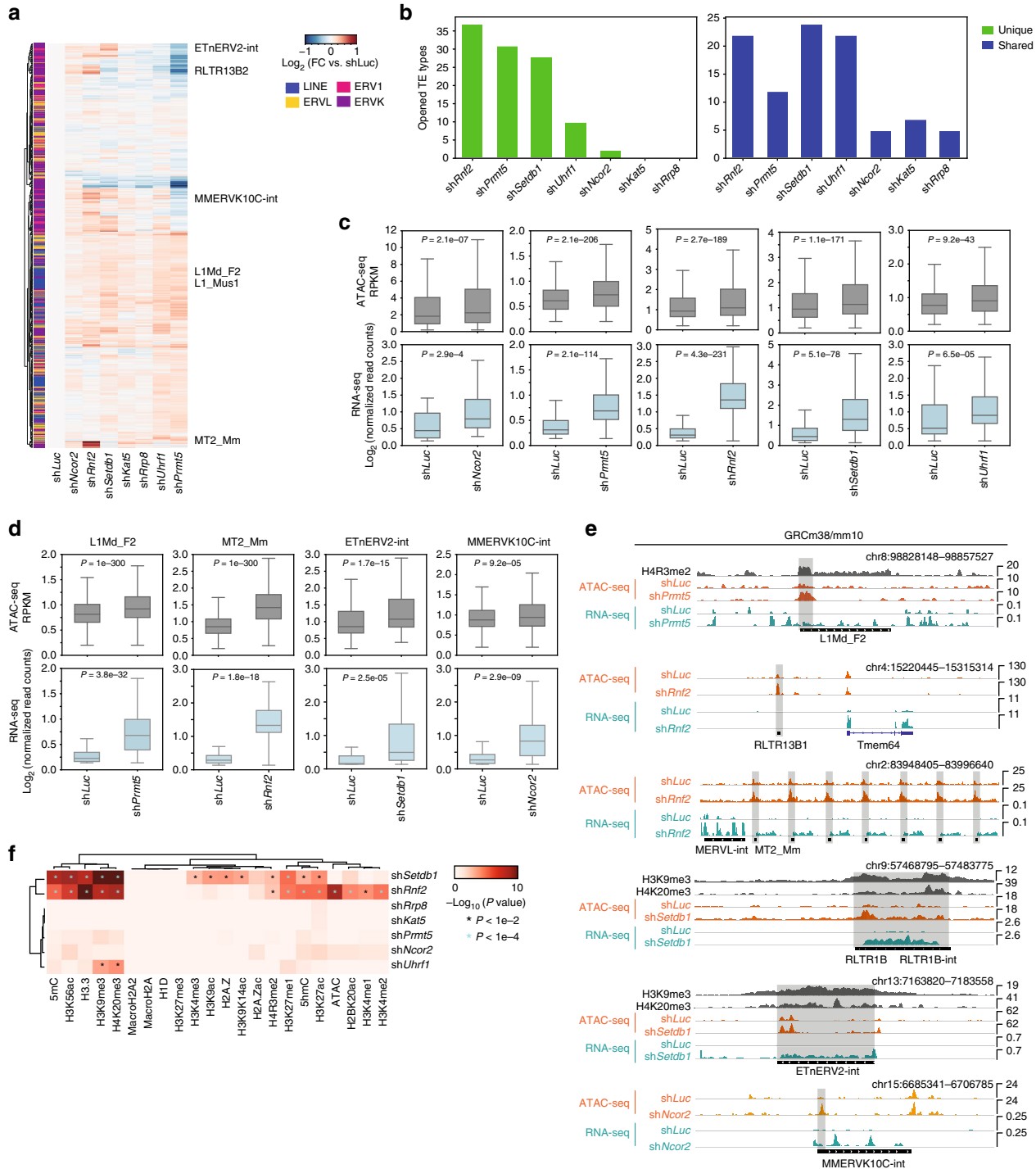

**Fig. 4** Loss of CMs opens chromatin at specific TEs. **a** Heatmap of the fold-changes in chromatin accessibility (measured by ATAC-seq) at TE types. The list of TE types are those that have 20% more or 20% less open or closed chromatin. Selected TE types are indicated, and the full table of all TE chromatin changes is in Supplementary Data 5. **b** Bar chart of the number of TE types that acquire open chromatin in a unique shRNA knockdown (left), or in more than one shRNA knockdown (right). **c** Boxplots showing the relationship between chromatin (ATAC-seq, top rows) and expression (RNA-seq, bottom rows). The boxplots contain all TE types that acquire open chromatin, for the selected shRNA knockdowns. $P$ values were derived from a Mann–Whitney $U$ test. **d** Boxplots of the levels of ATAC-seq (top rows, dark gray) and RNA-seq (bottom rows, light gray) for selected TE families for the indicated CM knockdown, for the indicated type of TE. $P$ values were from a Mann–Whitney $U$ test. **e** Example genome views of individual genomic TEs showing specific chromatin marks, expression (RNA-seq), and chromatin accessibility (ATAC-seq). The examples show TEs that acquire open chromatin and expression in the indicated shRNA knockdown. Examples include the LINE1 elements (L1Md_T, L1Md_A), ERVL (MERVL-int), MT2_Mm, ERVKs (RLTR13B1, ETnERV2-int, and MMERVK10C-int). **f** Enrichment of chromatin marks at TE types that become open in response to the indicated shRNA knockdown. Significance is derived from a Fisher exact test, $P < 0.01$ and is indicated with an asterisk. For the boxplots, the midline indicates the median, boxes indicate the upper and lower quartiles and the whiskers indicate 1.5*interquartile range

H4R3me2 (Supplementary Figure 1d). A similar relationship was observed for the H3K9me3 enriched ETnERV2-int which acquired open chromatin and RNA expression when *Setdb1* was knocked down (Fig. 4d, e). To explore these relationships, we measured the correlation between the chromatin marks and changes in accessibility (Fig. 4f). The clearest relationships were between sh*Setdb1* and heterochromatin marks H3K9me3 and H4K20me3, and other repressive marks, H3.3 and 5mC (Fig. 4f). Knockdown of *Rnf2* also correlated with repressive epigenetic marks (Fig. 4f). Overall, these results demonstrate that the loss of CMs leads to the alteration of chromatin at TEs, and a matching upregulation of the expression of TEs. There is a relationship between the CMs target, and the chromatin modifications marking the TEs, although due to the nature of CMs as members of multi-protein complexes, a direct relationship was not always observable.

**Knockdown of CMs activates early embryonic gene expression.** TEs are expressed and their chromatin is regulated in a phase-specific pattern in the early embryo[18,58]. TE expression has become a useful marker for early embryonic gene expression

programs in ESCs, as MERVL expression marks a small sub-population of 2C-like cells, that have totipotent properties[26], a capability ESCs lack. Knockdown of CMs can increase the fraction of these cells[15,59], but the relationship between CM, chromatin, MERVLs, and the 2C gene expression program remains unclear.

To shed light on the regulatory network controlling 2C-like cells, we interrogated our shRNA knockdowns for activation of a 2C-like gene expression program. Downregulation of *Rnf2*, *Brd7*, and *Hdac5* led to the upregulation of MERVLs, and a corresponding increase in 2C-stage embryonic genes, such as *Zscan4c* (Fig. 5a, b). To assess the type of embryonic gene program being activated we measured the expression of the significantly upregulated genes in each of the knockdowns, against RNA-seq data from a range of embryonic stages[60]. Genes upregulated in the *Rnf2*, *Hdac5*, and *Brd7* knockdowns were significantly highly expressed in 2C embryos, but not in other embryonic stages (Fig. 5c). Knockdown of *Setdb1* did not have this effect (Fig. 5c). This pattern could be confirmed by measuring the expression level of the top 200 upregulated genes (versus ESCs) in 2C embryos (Fig. 5d). A characteristic of the 2C stage is

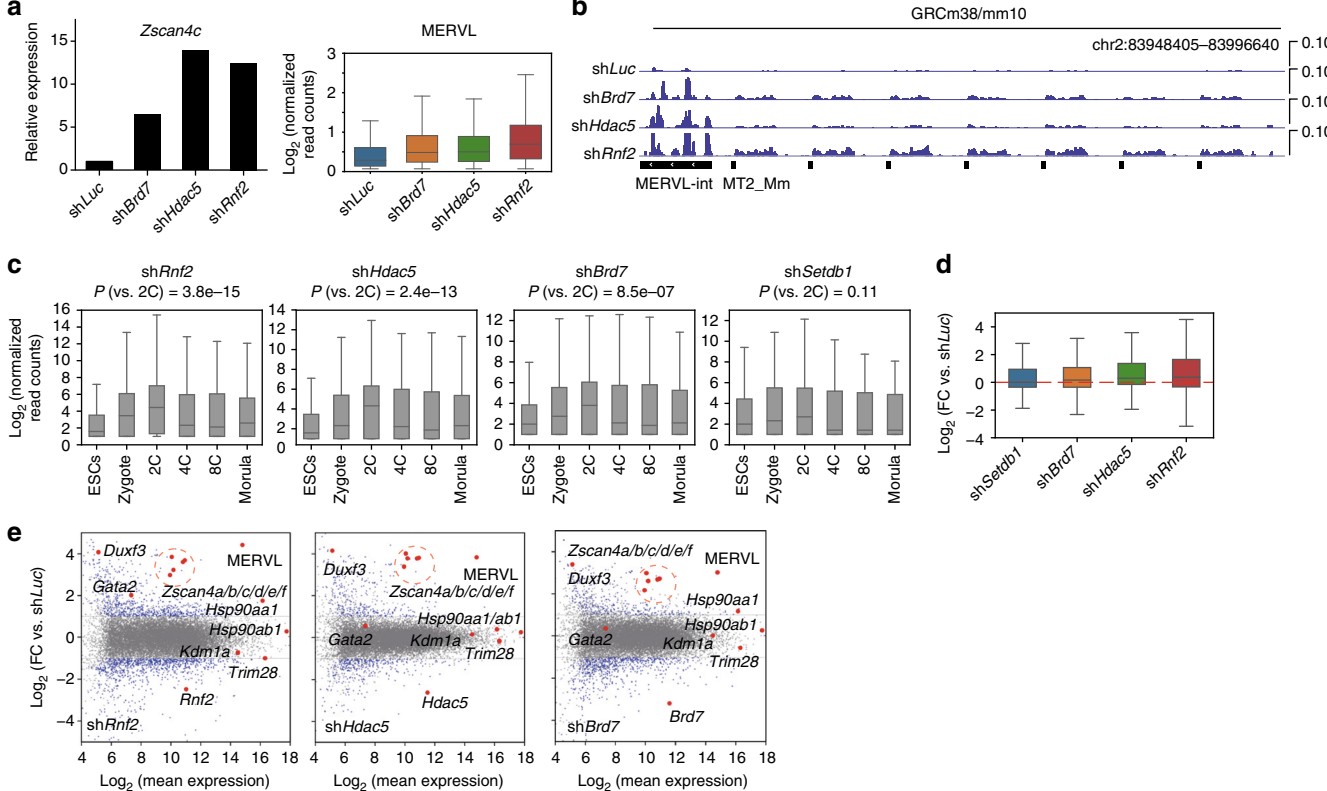

**Fig. 5** Knockdown of chromatin modifiers activates a 2C-like gene expression program. **a** RNA-seq expression level of the 2C-related gene *Zscan4c* in the indicated knockdowns (left), and a boxplot showing the level of MERVL TEs in the same knockdown RNA-seq data. Bar chart values indicate the mean of the two biological replicates (sh*Hdac5* or sh*Rnf2*), or a biological singlicate (sh*Brd7*), relative to sh*Luc* control knockdown. The boxplot shows the Log2 normalized read count mapping to the total set of MERVL TEs, from the same RNA-seq data. **b** Example genome view showing the RNA-seq pileup data and activation of MERVL-int and MT2_Mm TE expression in *Brd7*, *Hdac5*, and *Rnf2* shRNA knockdowns. **c** Boxplot of RNA-seq expression for genes upregulated in the indicated knockdowns. For the sets of genes significantly upregulated in the indicated CM knockdown, their expression was measured in the indicated embryonic stages and in ESCs (RNA-seq data was taken from[60]). *P* values are from a Mann–Whitney *U* test. The *x*-axis indicates the overall level of expression of the knockdown-specific genes, in those cell types. **d** Upregulation of a 2C-related gene signature by the indicated CM knockdowns. A 2C-related gene signature was generated by taking the fold-change of all 2C genes versus ESCs and taking the top 200 fold-change upregulated genes. The expression of those top 200 2C genes was then measured in the indicated knockdowns, relative to sh*Luc* control. **e** MA-scatter plots showing the expression of all genes and TEs in the sh*Rnf2*, sh*Hdac5*, and sh*Brd7* knockdowns. Labels and red spots indicate the expression of the shRNA target gene, the MERVL-int TE, and selected 2C-related genes (*Zscan4a/b/c/d/e/f*-cluster, *Duxf3*, *Hsp90aa1*, *Hsp90ab1*, and *Gata2*). Note that the effect is not related to deregulation of *Trim28* or *Kdm1a*, two known MERVL regulators. For the boxplots in this figure, the midline indicates the median, boxes indicate the upper and lower quartiles and the whiskers indicate 1.5*interquartile range

the splicing of MERVL TEs into genes on the same strand orientation[61], a pattern we observed only in the *Brd7*, *Hdac5*, and *Rnf2* knockdowns (Supplementary Figure 7a, b). These results showed that the 2C gene expression program was only seen in the sh*Brd7*, sh*Hdac5* and sh*Rnf2* knockdowns (Fig. 5d).

We next sought to understand how these knockdowns were activating a 2C gene expression program. Our analysis indicates that MERVLs are enriched for the activatory H3K56ac chromatin mark, but are not enriched for H2AK119ub (Fig. 1a and Supplementary Data 2). This is interesting as RNF2 catalyses the deposition of H2AK119ub, and BRD7 and HDAC5 can both target histone acetylation. These observations suggest two pathways for the activation of the 2C gene expression program; one direct, and one indirect, pathway. The direct pathway involved the acetylation of histones, as knockdown of the histone deacetylase *Hdac5*, and knockdown of the histone acetylation binder *Brd7* could upregulate MERVLs and 2C genes (Fig. 5a, c). Conversely, upregulation of MERVLs in the *Rnf2* knockdown, suggests an indirect pathway, as MERVLs are not enriched with H2AK119ub, but downregulation of H2AK119ub is a requirement for 2C-like cells[62]. The TF GATA2 has been implicated in driving cells to a 2C-fate[61], and in our data only knockdown of *Rnf2* could activate *Gata2* (Fig. 5e). Interrogating ChIP-seq data for RNF2 and H2AK119ub[63], indicated that they are both enriched at the *Gata2* locus (Supplementary Figure 7c). These results suggest that 2C-like cells can emerge by the manipulation of two pathways, one pathway involving H3K56ac, HDAC5, and BRD7, and an indirect pathway involving H2AK119ub, RNF2, and *Gata2*. These pathways complement other factors known to activate the 2C-like state, including *Chaf1a/b*, *Chd5*, *Rif1*, *Kdm1a*, *Ehmt2*, *Yy1*, *Rybp*, and miR-34a[15,26,28,61,64]. These results suggest that there are multiple convergent pathways that can drive cells to a 2C state, although whether all of these factors are converging on the same pathway or act independently remains unclear.

## Discussion
TEs are the single most prevalent genomic element, more numerous than coding genes or other regulatory units. Although potentially dangerous for genome integrity they are nonetheless a pool of potential regulatory sequences, chromatin control elements, and protein coding genes, and they can function as a substrate for evolutionary processes[3,9,21]. During early embryonic development there is extensive chromatin remodeling[58], and the activation of TE expression[18]. Consequently, a careful balance is maintained between the potential benefits of TEs, versus their deleterious aspects, and this control is carried out by the epigenetic regulatory system.

Heterochromatin formation is a major mechanism for the suppression of TEs, and we find that many TEs are marked by heterochromatic histone modifications. However, data suggests that, at least for LINEs and IAP-family TEs, TEs are repressed by the loss of activatory marks, rather than through the gain of repressive heterochromatin[19]. Our data indicate that TEs are marked by many activatory marks, like histone acetylations. We also observed bivalent marking of TE chromatin by H3K9me3 and H3K27ac, which may help to explain why we only observed a limited number of downregulated TEs in response to CM knockdown. This was surprising, as ESCs express over 200 TE types at levels comparable to many genes. Bivalent marking may mean that these TEs are kept in partially activated and repressed states.

TEs are increasingly implicated in a wide-range of biological functions in embryonic cell types. It has long been known that there is reverse transcriptase activity (indicative of TE activity), in developing mouse embryos[65], and HERVK-derived viral-like particles have been observed in normal human embryos[66]. TEs are expressed in a stage-specific pattern in both human and mouse embryos[18,19], and their expression in ESCs is associated with alternative properties. For example, MERVL expression marks the 2C stage of embryonic development as well as 2C-like cells in ESCs[26], and the inhibition of MERVLs, or MERVL-containing lincRNAs, impairs the transition from the zygote to a 4C embryo stage[67,68]. LINE1 RNA is transiently required for 2C stage embryos to exit to the 4C stage, and in ESCs LINE1 is required for self-renewal, and chromatin remodeling[43,44]. It is increasingly clear that TEs are involved in normal biological processes.

In conclusion, our study reveals the complexity of epigenetic regulation of TEs in ESCs. TEs are marked not only by repressive chromatin marks, but are also widely marked by activatory chromatin marks, including patterns of simultaneous marking by activatory and repressive marks. We show that the loss of CMs caused specific changes in TE expression and chromatin accessibility, and these effects are not limited to the formation of heterochromatin or DNA methylation, but encompass a large fraction of known chromatin modifications. This work suggests that the regulation of TEs is a function of overlapping epigenetic pathways, and that a major task of the epigenetic system is to manage the expression of TEs.

## Methods
**Cell lines and cell culture medium**. All experiments were performed in mouse E14 ESCs (RRID:CVCL_C320), to avoid any differences in genetic background that may influence the number of TEs. Mouse ESCs were grown in serum + LIF medium: DMEM-HIGH GLUCOSE (SH30022.01, HyClone), 15% (v/v) fetal bovine serum (S1580–500, Biowest), 1% (v/v) L-Glutamine (35050-061, GIBCO), 1% (v/v) sodium pyruvate (25-000-CIR, Corning), 1% (v/v) non-essential amino acids (25-025-CI, Corning), 0.5% (v/v) penicillin-streptomycin solution (SV30010, HyClone), 0.1 mM 2-mercaptoethanol (21985-023, GIBCO), and 1000 U/ml mLIF (ESG1107, Millipore).

**Chromatin modifier knockdowns**. CMs were knocked down in serum + LIF medium using pLKO.puro lentiviral system as previously described[27,41]. Briefly, pLKO.puro lentiviruses were added to ESCs and the medium was changed after 8 h. Cells were selected with 2 μg/ml puromycin (Gibco) for 2 days, and harvested after a further 2 days. Sequences for the shRNAs are in Supplementary Data 6.

**Western blot**. Histone proteins were extracted by acid precipitation. Briefly, cells from a 6-well dish were scraped, washed with PBS and resuspended in extraction buffer (PBS, 0.5% Triton-X-100 (9002-93-1, Sigma-Aldrich), 5 mM sodium butyrate (B5887, Sigma-Aldrich)), and allowed to extract for 20 mins. The extract was centrifuged at 14,000× *g* for 20 mins and the supernatant was discarded. The pellet was resuspended in 0.2 N HCl and left at 4 °C overnight. The mixture was vortexed, and the pH was neutralized with 1 M Tris-HCl pH 8.0. Western blots were performed using typical laboratory procedures with the antibodies: anti-H3K9me3 (1:1000; ab8898, abcam), anti-H3K4me3 (1:1000; ab8580, abcam), anti-H3K4me1 (1:1000; ab8895, abcam), anti-H3K27me3 (1:1000; 07-449, Millipore), anti-H3ac (1:1000; 06-599, Millipore), anti-H4K12ac (1:1000; ab46983, abcam), anti-H4ac (1:1000; 06-886, Millipore), anti-H3K27ac (1:1000; ab4729, abcam), anti-H2AK119ub (1:1000; 8240, Cell Signaling Technology), anti-H3 (1:1000; ab1971, abcam), anti-OCT4 (1:10,000; SC-8628, Santa Cruz), anti-NANOG (1:1000; A300-397A, Bethyl), and anti-GAPDH (1:10,000; MAB374, Millipore). Uncropped raw images of the Western blot membranes are in Supplementary Figure 8.

**RNA-seq and RT-qPCR experimental protocols**. RNA-seq was performed as described[69]. Briefly, a 6-well dish of ESCs was scraped and lysed in TRIzol (RN190-200, Invitrogen). RNA was extracted according to the manufacturer's instructions, and 1 μg was submitted for sequencing. All RNA-seq was sequenced to a depth of 50 M tags, and 150 bp paired-end. For RT-qPCR RNA was extracted using TRIzol (RN190-200, Invitrogen), cDNA was prepared with reverse transcriptase kit (RR36B, Takara), and RT-qPCR was performed with SYBR Green Master kit (RR820B, Takara). RT-qPCR primers are described in Supplementary Data 7.

**ATAC-seq and native-ChIP-qPCR experimental protocols**. ATAC-seq was performed as described[57,70]. A total of ~50,000 cells were washed once with 50 μl of cold PBS and resuspended in 50 μl lysis buffer (10 mM Tris-HCl pH 7.4, 10 mM NaCl, 3 mM MgCl2, 0.2% (v/v) IGEPAL CA-630). The suspension of nuclei was then centrifuged for 10 min at 500 × *g* at 4 °C, followed by the addition of 50 μl

transposition reaction mix (25 μl TD buffer, 2.5 μl Tn5 transposase and 22.5 μl nuclease-free $H_2O$) of Nextera DNA library Preparation Kit (96 samples) (FC-121-1031, Illumina). Samples were then PCR amplified and incubated at 37 °C for 30 min. DNA was isolated using a MinElute Kit (Qiagen). ATAC-seq libraries were first subjected to five cycles of pre-amplification. To determine the suitable number of cycles required for the second round of PCR, the library was assessed by quantitative PCR[70], and the library was then PCR amplified for the appropriate number of cycles. Libraries were purified with a Qiaquick PCR (Qiagen) column and were quantified using a KAPA Library Quantification kit (KK4824, Kapa Biosystems) according to the manufacturer's instructions. Library integrity was checked by gel electrophoresis. Finally, the ATAC library was sequenced on a NextSeq 500 using a NextSeq 500 High Output Kit v2 (150 cycles) (FC-404-2002, Illumina) according to the manufacturer's instructions.

Native-ChIP-qPCR was performed as described[71]. Ten million cells were centrifuged and resuspended in Buffer 1 (0.32 M Sucrose, 60 mM KCl, 15 mM NaCl, 5 mM $MgCl_2$, 0.1 mM EGTA, 15 mM Tris-HCl pH 7.5, 0.5 mM DTT, 0.1 mM PMSF, 1:1000 protease inhibitor cocktail (Sigma-Aldrich), along with 2 ml of Buffer1 + 0.1% IGEPAL. The resulting 4 ml of nuclei were layered on top of 8 ml of Buffer 3 (Same as Buffer 1, but with 1.2 M Sucrose) and centrifuged at $10,000 \times g$ for 20 min at 4 °C unbraked. Nuclei were resuspended in Buffer A (0.34 M sucrose, 15 mM Hepes [pH 7.4], 15 mM NaCl, 60 mM KCl, 4 mM MgCl2, 1 mM DTT, 0.1 mM PMSF, 1:1000 protease inhibitor cocktail (Sigma-Aldrich)) and digested for 10 mins at 37 °C with MNase (Affymetrix) in buffer A supplemented with 3 mM $CaCl_2$. The reaction was stopped with 5 mM EGTA, centrifuged at $13,500 \times g$ for 10 min, and chromatin resuspended in (10 mM EDTA [pH 8.0], 1 mM PMSF, 1:1000 protease inhibitor cocktail) and rotated at 4 °C for 2–4 h. The mixture was adjusted to 500 mM NaCl, allowed to rotate for another 45 min and then centrifuged at $13,500 \times g$ for 10 mins. Chromatin supernatant was diluted to 100 ng/μl with buffer B (20 mM Tris [pH 8.0], 5 mM EDTA, 500 mM NaCl, 0.2% Tween 20) and incubated for 20 mins at 4 °C with 60 μl protein G beads (GE Healthcare). Antibody was added and rotated overnight at 4 °C. A volume of 100 μl of Protein G beads were added and the sample was rotated for a further 3 h at 4 °C. The beads were washed three times with buffer B, and once with buffer B without Tween 20. The DNA was eluted with 300 μl of elution buffer (20 mM Tris [pH 7.5], 20 mM EDTA, 0.5% SDS, 500 μg/ml Proteinase K) and incubated for 4 h at 56 °C. The resulting samples were extracted with phenol-chloroform followed by purification using a Qiagen MinElute columns, according to the manufacturer's instructions. Antibodies used: anti-H3K9me3 (5 μg; ab8898, abcam), anti-H3K27ac (5 μg; ab4729, abcam). qPCR was performed with SYBR Green Master kit (RR820B, Takara). Primer pairs are described in Supplementary Data 7.

**RNA-seq and TE RNA-seq computational analysis**. As a quality control step the RNA-seq data was analyzed for normal gene expression, using the method described[60,69]. Briefly, reads were aligned to the genome using bowtie2[72] and RSEM[73], with the settings '—bowtie2—bowtie2-sensitivity-level very_sensitive—no-bam-output—estimate-rspd' using an index built against the ENSEMBL v81 transcriptome, and normalized using EDAseq[74] (v2.4.1) (which = 'full'). Samples that did not closely correlate with other replicate samples, clustered closer to unrelated samples, or were outliers ($R^2$ correlation < 0.6), were deleted from the analysis.

For the analysis of TEs, reads were mapped to the mouse genome (mm10) using the STAR aligner[75], with the options:—readFilesCommand zcat—outFilterMultimapNmax 100—winAnchorMultimapNmax 100—outMultimapperOrder Random—runRNGseed 777—outSAMmultNmax 1—outSAMtype BAM—outFilterType BySJout—alignSJoverhangMin 8—alignSJDBoverhangMin 1—outFilterMismatchNmax 999—alignIntronMin 20—alignIntronMax 1000000—alignMatesGapMax 1000000. Tags mapping to genes and TEs were counted with the option:—mode multi. GTF files for gene annotation were downloaded from GENCODE (M9). GTF files for TE annotations were downloaded from the TEtranscripts website (mm10_rmsk_TE.gtf.gz). For the quantification of genes and TEs, samples were normalized using EDAseq[74] (v2.4.1) (which = 'full'). The GC content for TEs was generated by taking the average for all TEs. TEs were normalized at the same time as the genes. This has the effect of suppressing the magnitude of the fold-changes, however the fold-changes reported here are similar to those reported in other studies that used knockdowns[40]. Differential expression was called using DESeq2[75], with a P value < 0.05 (Benjamini-Hochberg-corrected) and an absolute fold-change > 2. Gene ontology analysis was performed using goseq (v3.2.5)[77]. Other analysis was performed using glbase[78].

The age of TEs was calculated as in[8,11]. Briefly, the average divergence between the TE repeats and the consensus (milliDiv from RepeatMasker UCSC track) was calculated for each TE type, and the Jukes-Cantor method with a substitution rate of 4.5e-9 per base per year was applied. This gave an estimate for TE age in years. The results were broadly consistent with other estimates of TE age[8,11].

The list of 2C-related genes was generated by calculating the fold-change of 2C genes versus ESC genes, and taking the top 200 2C-associated genes. RNA-seq data was from[60]. A second approach was used to generate a 2C-associated gene signature, by taking the top 200 upregulated genes in each CM knockdown, and then measuring the fold-change of those genes in the 2C sample, relative to the ESCs.

MERVL-associated genes were defined as those genes that have a MERVL (either MERVL-int or solo-LTR MT2_Mm) within 10 kb up- or downstream of the gene body. A gene was annotated as the same orientation (same ori.) if it had a MERVL in the same strand flanking it, or in the opposite orientation (opposite ori.) if it had a MERVL element in the opposite strand, as described in[61].

Mapping statistics and details of all the samples used in this study are contained in Supplementary Data 1.

**ChIP-seq and ATAC-seq computational analysis**. ChIP-seq data for histone modifications, variant histones, DNA methylation, chromatin accessibility, and a selection of TFs and chromatin modifiers (Supplementary Data 1), were downloaded from GEO/SRA and the reads were aligned to the mouse genome (mm10) using bowtie2 (v2.2.5)[72] with the options: -p 20—very-sensitive—end-to-end—no-unal. To analyze repetitive sequences, only the best alignment is reported for multimapped reads, if more than one equivalent best alignment was found, then one random alignment was reported. Reads mapping to mitochondrial DNA or unassigned sequences were discarded. For pair-end sequence data, only concordantly aligned pairs were kept. Alignment bam files were transformed into read coverage files (bigwig format) using deepTools[79] with the RPKM (Reads Per Kilobase per Million mapped reads) normalization method. For analysis of TEs, coordinates and annotations of TEs were downloaded from the UCSC Genome Browser (GRCm38/mm10) version of RepeatMasker. TEs shorter than 300 bp or those TE types with less than 50 copies were deleted from the analysis. For TE enrichment analysis, as many ChIP and ATAC signals were not evenly distributed across TEs, we divided the TEs into evenly spaced 500 bp bins. The coverage signal on each bin was extracted using deeptools, and the bin with the maximum signal was used as the observed value. For the expected background value, random genomic regions of the same number as the TE type and the same size as the TE were taken, the coverage signal was measured, and the maximal bin was taken as the randomly expected background. Enrichment was calculated by taking the observed over the expected enrichment, and is expressed as log2(fold-enrichment). Quality control was performed by measuring the reads per kilobase of TE per million library sequence reads (RPKM) at all TEs, and then combining replicates where the Pearson correlation was >0.6 (Supplementary Figure 1a). Outlier samples that did not correlate closely with other ChIP-seq data samples of the same type were deleted from the analysis. The full data set of ChIP-seq experiments used in this study is detailed in Supplementary Data 1. Heatmaps and pileups were generated using deepTools[79], motif discovery was performed using MEME[80].

**Quantification and statistical analysis**. No statistical methods were used to predetermine sample size, and experiments were not randomized. The investigators were not blinded to the allocation of experiments, nor the outcome. The number of biological replicates are described in the figure legends, or in Supplementary Data 1. Samples were considered statistically significant at $P < 0.05$, unless indicated. The statistical test used is indicated in the figure legend.

**Reporting Summary**. Further information on experimental design is available in the Nature Research Reporting Summary linked to this article.

**Data availability**
The datasets supporting the conclusions of this article are available in the Gene Expression Omnibus (GEO), under accession number GSE108091. The authors declare that the data supporting the findings of this study are available within the article and its Supplementary Information files, or from the corresponding author upon reasonable request. A Reporting Summary for this Article is available as a Supplementary Information file. The Source Data underlying Figs. 1d, 3h, and Supplementary Figs 2h and 5b are provided as a Source Data file.

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

## Acknowledgements

We thank Ralf Jauch for discussions. This work was supported by the National Natural Science Foundation of China (31801217, 31850410463, 31850410486, 31671537, 31501185), Shenzhen Peacock plan, the Shenzhen Science and Technology Innovation Committee general program (JCYJ20170307110638890), and the Science and Technology Planning Project of Guangdong Province (2017B030314056).

## Author contributions

J.H. performed the bioinformatic analysis, and prepared the manuscript. X.F. did the ATAC-seq, RNA-seq, M.Z., W.L., M.M.A., J.Z. performed the knockdown panel RNA-seq. H.L., W.K., F.H., C.C. performed experiments, L.S., Y.L., I.A.B., helped with the bioinformatics. A.P.H. conceived the project, supervised and funded the study, and prepared the manuscript with assistance from M.A.E, C.J, Q.Z., I.A.B, and Y.-H.L.

## Additional information

**Competing interests:** The authors declare no competing interests.

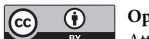

