## [Peer Review File · Nature Communications]

Reviewer #1 (Remarks to the Author):

He et al present a genome-wide study of how transposable elements (TEs) are regulated by chromatin modifications, and chromatin-modifying enzymes, in mouse embryonic stem cells. The mechanisms by which TEs are regulated during early development is of wide interest in developmental biology, regenerative medicine and genomics, and many open questions remain. In particular, what are the mechanisms by which TEs are controlled? How generic or not are these mechanisms across TE types? How do TEs actively contribute to cell states and transitions?

This work represents an important, fundamental advance in the field. It is ambitious in its scale, tackling the entirety of TEs, dozens of chromatin marks. The data produced will be an invaluable resource, of which none presently exists. The main finding, that TEs are marked by complex and specific histone marks, many of which are activating (not repressive) is an important finding.

The paper is well written (although use of commas is notably lacking, particularly in Abstract). The figures are attractive and informative. I detect no major conceptual or methodological weaknesses. Nevertheless there are a number of issues that are raised by the paper, often because the scale and fundamental nature of the work raises many interesting questions for follow up work. Therefore, several of the comments below might fall outside the scope of the paper, but nevertheless are interesting to mention.

1) A main concern when working with TEs, is their repetitive nature and consequent difficulty in mapping NGS reads to them, and ambiguity as to precise origin of TE-mapping reads, or even the discarding of such reads due to multi-mapping. Could the authors comment on: how this is dealt with in the analysis / to what extent this affects (or not) this work and conclusions arising from it / what are possible artefacts in the data arising from these effects? This could be mentioned also in the main manuscript, even if briefly.

2) Fig 1A and others: I am missing well-known TEs like B1, B2, MIR, L2 in this Figure and others. Were they included in analysis? Could they be indicated in the relevant figures?

3) Fig 1B and others: A big complication of working with TE annotations, is that most TE instances represent a fragment of the full TE consensus. It is not clear

how this is dealt with in the numerous "metagene" analyses throughout the paper. Are TEs aligned by consensus, or not? If not, then I would expect that this would introduce serious "blurring" into the metagene profiles, since the TEs will not be aligned by their sequence. It would be nice to see a

Supplementary Figure, or several, showing histone modification/CM profiles, but explicitly profiled across the full TE consensus and normalising for the insertion frequency of each TE consensus nucleotide (because not every nucleotide of the consensus is included in 100% of inserted instances of a TE).

4) The authors don't discuss much how consistent are histone modification/ CM binding between the multiple instances of the same TE type. Presumably some TEs are highly consistent (eg Fig1C) while others are not. One might expect this heterogeneity is correlated with TE age, since highly divergent sequences might give rise to differences in TF binding. Could this heterogeneity be quantified?

5) One notable omission is lack of citation for the transcription factor REST / NRSF. This was one of the first TF-TE interactions discovered genome-wide (Johnson et al, NAR 2006) and predates the cited papers on Line139. Please cite this, and also, indicate on Figure 2A which TEs REST is here binding to, and whether this supports previous findings.

6) Something else lacking in the paper is any discussion of TE ages. We know that these span a huge timespan, and that some remain active while others are ancient fossils. I request the authors mention this, and at least consider working this consideration into their present analyses. For example, are recent TEs more likely to be expressed in RNA, compared to ancient ones? Do ancient ones consequently have different types of chromatin modifications, perhaps more active ones? Are ancient (and therefore, benign) TEs more likely to be exapted to help with gene regulatory processes compared to new (and more dangerous) ones?

7) The analysis of TE expression is very interesting, and yet it seems a little under-developed. What % of instances of each TE type are detected as RNA? Similar to above, are some TEs very uniform in their expression, while others are not? Are incomplete TE insertions less likely to be expressed compared to intact ones? Do TEs with activating marks have higher RNA production?

8) Given the discovered relationship between TEs / CMs / transcription factors, can the authors identify in some cases the motif for those TFs within the TE consensus, as done by Johnson et al and Bourque et al ?

9) An interesting experiment suggested by this work, would be to knock down or overexpress candidate TE RNAs to promote the acquisition of the 2C state in ESCs.

10) One impactful conclusion of this work, is that the idea that TEs are uniformly marked by repressive histone modifications is outdated. This might be presented more forcefully in the Abstract and Conclusions.

Minor issues:

- What is y axis of Fig5C? Unclear at present.

-Please show genome assembly version together with coordinates, in appropriate Figures.

-Figure S5 - What are the columns? What does the red line represent? Legend is not informative here.

-Fig 1B and others: Where appropriate, it would help to display the n number representing the number of unique TE instances that each figure is displaying.

-Line 323 - numerous, not numerical.

-Line 441 - text missing.

Reviewer #2 (Remarks to the Author):

He et al have performed a large-scale data mining effort to identify chromatin signatures associated with TEs in mouse ES cells, finding a variety of profiles across different subfamilies. Knockdown of a panel of TE-associated chromatin modifiers uncovered class-specific regulators of TE transcription.

This study is most useful in summarising the TE epigenomic landscape and in providing a useful resource in the form of ATAC-seq and RNA-seq data from knockdowns of key TE regulators. What is harder to pick out is which novel results have been uncovered, as many of the highlighted findings are already known, such as: 1) the regulation of L1s and IAPs by PRDM5-mediated arginine methylation, 2) the enhancer profile of RLTR13D6 and other elements, 3) the TE classes that are regulated by SETDB1. That the emerging patterns are complex and context-specific is also not a novel insight, as the independent evolutionary paths of different TEs, along with their age, are known to lead to these different epigenetic profiles. Nevertheless, I appreciate the strength of combining data mining with the mini knockdown screen, and could see this work becoming an important reference point for anyone interested in TE epigenetics.

Technically, however, the study suffers from major flaws in the analysis of ChIP-seq data:

1) Non-uniquely mapped reads were included in all analyses. Whilst this was not spelled out in the methods section, it is clear from the profiles displayed throughout, which do not have the expected decreased mappability across young TEs. The default output from bowtie2 will assign reads with multiple hits of equal quality to one of those locations at random. Therefore, mapping to individual TE copies is ambiguous and no claims can be made about which copies are enriched in which marks. This sort of mapping is only acceptable when generating class-wide profiles, such as those in Figure 1b. In contrast, profiles that display individual TE copies (e.g., heatmaps in Figure 1c and browser snapshots in Figure 1d) are inaccurate and misleading. To demonstrate the difference, I have remapped the H4R3me2 data and extracted uniquely mapped reads. In the attached file I show the profile for the L1Md_T element displayed in Figure 1d (see panel A) and the heatmap for L1Md_F3 elements from Figure 1c (see panel B). Importantly, this has a major impact on any classification that relies on the % of copies that bear a given epigenetic mark, which the authors use throughout. To use this classifier the authors have to: a) use uniquely aligned reads, and b) work out the % only relative to copies that have sufficient mappability to make the claim that a given epigenetic mark is present/absent. This will inevitably affect younger elements more than older ones. Claims that RLTR13D6 TEs bear both activating and repressive marks at the same copies are also potentially affected by this.

2) No peak calling algorithm was used. Even though it is missing from the methods section, the legend to Figure 1 states that TEs were classified as marked based on a cut-off of reads per million (but not normalized to TE length?). The problem with this approach is that it dampens the signal from localized peaks within TEs and does not take into account the background from an input sample. For example, the 5' UTR of L1 elements harbours many peaks for proteins associated with its regulation (see, e.g., PMID 29802231). Panel C of the attached figure shows how 5hmC (from the same dataset used by the authors) is highly enriched at full-length L1Md_T elements, yet was not picked up by the authors as an enriched mark. I also would have expected to see enrichment for at least H3K4me3 and H3K27ac at the same elements. Marking of a given TE copy should be based on peak calling, using the appropriate input samples. The ATAC-seq data generated by the authors are

an exception and did involve peak calling, but it looks again like this was based on non-uniquely mapped reads.

3) Full-length and truncated elements are analysed together. One cannot generalize the pattern of a TE class when full-length elements are pooled together with much shorter elements that bear only part of the coding or regulatory regions of that TE. This affects L1 elements most prominently, as the vast majority of truncated elements lacks the regulatory 5'UTR. Therefore, 5hmC enrichment can be seen at full-length elements, but not truncated ones (see panel C of the attached figure), and the same will be true of other marks associated with the 5' UTR. This is less of an issue with LTR elements, as the Repeatmasker annotation separates LTR ends from the coding region.

The manuscript therefore needs a major reshuffle of its bioinformatic approaches, to avoid becoming a minefield of artefacts associated with the complexity of analyzing short sequencing data from TEs.

Reviewer #3 (Remarks to the Author):

This manuscript provides data relevant to mechanisms involved in the substantial context-specificity of the chromatin modification system, which marks and regulates TE using a regulatory chromatin code. The authors showed that TEs are marked not only by heterochromatic marks, but are labelled by all major types of chromatin modification in complex patterns in mESC. They knocked-down 41 chromatin modifiers and found 29 significantly deregulated at least one type of TE. Significant effect was found in the loss of Setdb1, Ncor2, Rnf2, Hdac5, Prmt5, Kat5, Uhrf1, Rrp8 and Ash2l, which caused widespread changes in TE expression and matching changes in chromatin opening. These effects were context-specific, as different chromatin modifiers regulated the expression and chromatin accessibility of specific subsets of TEs. Some of the content are relative novel, however, the manuscript covers many problems and needed to be largely and carefully improved.

Major points:

1. The novelty is moderate. The epigenetic regulation of transcription, and the relationship between the chromatin marks and the chromatin factors are well known and not novel at all.

2. The functional validation of TE regulation is lacked. Does the changed of TEs caused by CMs affect genome stability?
3. The information in the introduction is too simple and incomplete. The information about TE function, regulatory mechanism and TE-related diseases are not included. The previous publications about the relationship between TE and epigenetic regulation are not described. The current achievement and key issues needed to be solved are not clear.
4. This manuscript shown that TEs are also marked by active chromatin marks, such as H3K4me1, and H3K4me3. Moreover, the expression of the TEs are down regulated when CMs are knocked down. The regulatory mechanisms of these TEs have not been studied and explained. Are they also suppressed in ES cells?
5. As shown in the manuscript, in agreement with the prevalence of chromatin marks at TEs, CMs were also often bound to TEs in specific patterns. Is the pattern of CMs consistent with chromatin markers? The manuscript gives a few examples, however, the global analysis is lacking.
6. To rule out the change of cell fate in response to CM KD, the detection of pluripotency and differentiation gene expression is not enough. Protein expression level and ES cell phenotype should be detected.
7. To prove the relationship between TE expression and chromatin markers, or the relationship between the change of chromatin markers and chromatin state after the depletion of CMs, there is a lack of global wide statistics. The current data are difficult to explain it adequately, because the examples given are different in two parts.
8. In the part of that chromatin modifiers alter chromatin accessibility at specific sets of TEs, the TEs with up-regulated expression and opened chromatin are described. It is necessary to describe the TEs with down-regulated expression and closed chromatin.
9. “we chose to KD over a short time period of 4 days.....” at page 8. 4 days after KD is the standard time point to check ESC differentiation. The authors should provide the reasons and evidence why they chose the time point.

Minor points:

1. The manuscript could be written in a more concise manner to transmit clearer its important message. The English should be largely revised, for example at page 2, potential danger to genomic stability or instability? Page 3. Cell's DNA should be instead by cellular genome. Page 10, “although” and “nonetheless” should not coexist in one sentence. Page 11, “looking at” is so informal in manuscript.....I hope the author should pay more attention to improve the whole English grammar and the verbal accuracy, not only revise what I mentioned here.

2. There are many sentences with grammar mistake, which need to be carefully revised by the author. Such as, page 2, “particularly potent” is an adjective and cannot be used as subject. Page 7, “As many of the IAPs had methylated DNA (Fig.1c), and, as previously reported, these IAPs were bound by MBD-family proteins.” In page 7, “also” appeared twice in the next sentence. The meaning of “Overall the RLTR46.....of the TE RLTR46 copies” is confused. Page 13, “of the KDs,” also has serious grammar mistake. Page 14, “Although whether all of these pathways...”. “although” cannot be used in the front of an independent sentence. The author should also pay attention to the repeated use of words. Word’s diversity should be encouraged. Page 7, “as previously reported” appeared twice nearby.

3. In page 11, the author only provides examples that KD specific CM would open chromatin and up-regulate corresponding TEs. Does it mean all CMs negatively regulate TEs and maintain compact chromatin? If not, the author should also give opposite examples.

4. Page 14, it is improper to say that “Chaf1a/b, Chd5, Rnf2, Rif1, Kdm1a, Ehmt2, Yy1, Rybp or miR-34a” are pathways.

5. The author should pay attention to gene versus protein terminology.

6. Line 168, the protein pol II should be written as Pol II.

7. Page 3 line 49: A correct punctuation mark is missed.

8. Fig.1C, the authors should provide the color bar and scale bar, as well as for other heatmaps.

9. Fig1D and 3E, the authors should provide the range value of histogram.

10. Figure legend

10.1 For Fig.2, some of the information was not complete.

(a) The author should mark the X- and Y-axis.

(b) Epigenetic marks are pink.

5.2 For fig.S4b and S5a, “sh” should added in the each gene name at X-axis.

5.3 For fig.3b and 4a, the legends should be coincident with the figures. The Y-axis of fig.3b, FC or log₂ (FC)? And RPM or log₂ (RPM) for the label of fig.4a?

Reviewers' comments:

Reviewer #1 (Remarks to the Author):

He et al present a genome-wide study of how transposable elements (TEs) are regulated by chromatin modifications, and chromatin-modifying enzymes, in mouse embryonic stem cells. The mechanisms by which TEs are regulated during early development is of wide interest in developmental biology, regenerative medicine and genomics, and many open questions remain. In particular, what are the mechanisms by which TEs are controlled? How generic or not are these mechanisms across TE types? How do TEs actively contribute to cell states and transitions?

This work represents an important, fundamental advance in the field. It is ambitious in its scale, tackling the entirety of TEs, dozens of chromatin marks. The data produced will be an invaluable resource, of which none presently exists. The main finding, that TEs are marked by complex and specific histone marks, many of which are activating (not repressive) is an important finding.

The paper is well written (although use of commas is notably lacking, particularly in Abstract). The figures are attractive and informative. I detect no major conceptual or methodological weaknesses. Nevertheless there are a number of issues that are raised by the paper, often because the scale and fundamental nature of the work raises many interesting questions for follow up work. Therefore, several of the comments below might fall outside the scope of the paper, but nevertheless are interesting to mention.

1) A main concern when working with TEs, is their repetitive nature and consequent difficulty in mapping NGS reads to them, and ambiguity as to precise origin of TE-mapping reads, or even the discarding of such reads due to multi-mapping. Could the authors comment on: how this is dealt with in the analysis / to what extent this affects (or not) this work and conclusions arising from it / what are possible artefacts in the data arising from these effects? This could be mentioned also in the main manuscript, even if briefly.

Response: We thank reviewer for this question. We have added a small part to the results section that briefly mentions how we deal with multimapped reads (marked in red in the text). This section highlights to the reader how we consider TEs as metagenes, and how the analysis of individual TE copies can be ambiguous. See the reply to reviewer 2 below, which also deals with the issue of multimapping reads.

2) Fig 1A and others: I am missing well-known TEs like B1, B2, MIR, L2 in this Figure and others. Were they included in analysis? Could they be indicated in the relevant figures?

Response: For Figure 1a we only show those TEs with high levels of fold-enrichment. The full set of TEs used, including L2's is in Figure S1b and Supplementary Table 2. We found that the analysis of short <300 bp TEs was not accurate using for ChIP-seq data, using our methods. Hence B1, B2 and MIR SINEs were not included as we deleted TEs with a short average length. These TEs are included in the RNA-seq analysis, as with the longer reads we are more confident that we can detect them accurately. However, as we did not include them in the ChIP-seq analysis, we do not comment on them in the text. We have added this detail into the methods (marked in red in the text).

3) Fig 1B and others: A big complication of working with TE annotations, is that most TE instances represent a fragment of the full TE consensus. It is not clear how this is dealt with in the numerous "metagene" analyses throughout the paper. Are TEs aligned by consensus, or not? If not, then I would expect that this would introduce serious "blurring" into the metagene profiles, since the TEs will not be aligned by their sequence. It would be nice to see a Supplementary Figure, or several, showing histone modification/CM profiles, but explicitly profiled across the full TE consensus and normalising for the insertion frequency of each TE consensus nucleotide (because not every nucleotide of the consensus is included in 100% of inserted instances of a TE).

Response: We do not align the TEs by their consensus, and it does introduce some blurring of the TE copies. As the reviewer indicates, there are many ways to analyze TEs, and unfortunately each method introduces its own biases. We believe that the analysis of TEs as metagenes provides the best compromise between the various methods. This is particularly in light of reviewer 2's comments on the ambiguity in considering individual copies of TEs, and

working with multimapped reads. Instead, we have added a new section which describes differences in short truncated LINEs, and longer near intact LINEs. This TE type acts as a good example as it is relatively evolutionarily young, and we can clearly demonstrate how the chromatin marking the longer LINEs is quite different from the chromatin patterns at shorter LINEs (See below, and **Supplementary Fig. S1e, S1f**).

4) The authors don't discuss much how consistent are histone modification/ CM binding between the multiple instances of the same TE type. Presumably some TEs are highly consistent (eg Fig1C) while others are not. One might expect this heterogeneity is correlated with TE age, since highly divergent sequences might give rise to differences in TF binding. Could this heterogeneity be quantified?

Response: We are grateful for this comment, and we have added an analysis of TE age versus expression and chromatin marks into the paper (See below, comment 6). However, in response to reviewer 2, on the ambiguity in the analysis of individual TE copies, we have not explored the issue of TE age from the perspective of the individual copies.

5) One notable omission is lack of citation for the transcription factor REST / NRSF. This was one of the first TF-TE interactions discovered genome-wide (Johnson et al, NAR 2006) and predates the cited papers on Line139. Please cite this, and also, indicate on Figure 2A which TEs REST is here binding to, and whether this supports previous findings.

Response: We thank the reviewer for drawing our attention to this paper, and have cited it in the revision. The Johnson et al., paper indicates REST is binding to L2 LINEs. However, we did not find any evidence of REST binding to L2's in our analysis (**Supplementary Table 3**). An explanation for this discrepancy is that the Johnson et al study mainly looked at the human genome, whereas we used the mouse genome. L2 elements are more common in the human genome (~3.3%) versus the mouse (~0.3%) (Waterston et al., 2002), hence this may be a human-specific effect. We do see enrichment of REST on some ERVs (RMER21A, and RLTR44C), and some IAPs (IAPEY2_LTR, IAPEY3_LTR, and IAPEY_LTR). However, all of these TE types are predominantly found in the mouse genome and not the human genome. These results are in **Supplementary Table 3**, and we have added a plot of REST binding to RMER21A in **Supplementary Figure 2c**.

6) Something else lacking in the paper is any discussion of TE ages. We know that these span a huge timespan, and that some remain active while others are ancient fossils. I request the authors mention this, and at least consider working this consideration into their present analyses. For example, are recent TEs more likely to be expressed in RNA, compared to ancient ones? Do ancient ones consequently have different types of chromatin modifications, perhaps more active ones? Are ancient (and therefore, benign) TEs more likely to be exapted to help with gene regulatory processes compared to new (and more dangerous) ones?

Response: We are grateful for these suggestions. We have added a new column to **Supplementary Figure 1b** which shows that, in general, older TEs tend to lack chromatin marks, and younger TEs tend to be rich in chromatin marks, although there is much variation in this relationship. Additionally, we have also added **Supplementary Figure 5f**, which shows the correlation between TE age and fold-change in expression of KD. In general, this data suggests younger TEs tend to be more likely to be deregulated in the KDs. Although, once again, there is considerable variation and the correlation between TE age and fold-change is not strong.

7) The analysis of TE expression is very interesting, and yet it seems a little under-developed. What % of instances of each TE type are detected as RNA? Similar to above, are some TEs very uniform in their expression, while others are not? Are incomplete TE insertions less likely to be expressed compared to intact ones? Do TEs with activating marks have higher RNA production?

Response: We appreciate the reviewer for these questions. However, in response to criticisms raised by reviewer 2 we do not consider individual TE genomic elements, and no longer use the '% of TEs marked' as a measure. Consequently, these questions are now difficult for us to answer. Improvements in the quantitation of long-read sequencing data, and accurate sequencing across individual TE copies will be required to answer these questions.

8) Given the discovered relationship between TEs / CMs / transcription factors, can the authors identify in some cases the motif for those TFs within the TE consensus, as done by Johnson et al and Bourque et al?

Response: We have performed TE-wide motif discovery (**Supplementary Figure 2b**), however, we don't cover many TFs in this manuscript, as we focus on chromatin modifiers which mostly lack sequence-specific DNA binding domains (the list of chromatin modifiers was based on the Epifactors database, which sometimes overlaps with TFs (Medvedeva et al., 2015)). We briefly explore two CMs/TFs that have DNA-binding domains and were in our analysis, YY1 and REST. We detected binding of YY1 to IAPLTR2b and REST to RMER21A, and have added this to **Supplementary Figure 2c**.

In the course of this analysis, we found an interesting link between NR5A2 and RLTR13B. This link is interesting as NR5A2 is important in reprogramming primed ESCs (EpiSCs) to naïve ESCs (Guo and Smith, 2010). We use previously published ChIP-seq, RNA-seq and ATAC-seq data to show a strong correlation between NR5A2 binding at RLTR13B2 and RNA expression and chromatin opening in naïve ESCs (**Supplementary Figure 2e-i**). We would like to thank the reviewer for suggesting this idea, which adds a new aspect to our manuscript.

9) An interesting experiment suggested by this work, would be to knock down or overexpress candidate TE RNAs to promote the acquisition of the 2C state in ESCs.

Response: This is an excellent suggestion. However, it is technically challenging as traditional tools for manipulating genes do not work well when applied to TEs. shRNAs are notoriously poor at knocking down TEs, whilst plasmids suitable for efficient overexpression in ESCs are lentiviral-based and contain LTRs, which makes cloning other LTRs, or even ERVs inside these vectors inappropriate or even potentially dangerous. Consequently, this work requires the establishment of genome manipulation systems such as CRISPRa/CRISPRi, and is beyond the scope of the current work.

10) One impactful conclusion of this work, is that the idea that TEs are uniformly marked by repressive histone modifications is outdated. This might be presented more forcefully in the Abstract and Conclusions.

Response: We are grateful for this suggestion. We have added more stress to this in the revised manuscript, and have added new native-ChIP-qPCR data which shows that for a

select number of individual genomic TEs, they are simultaneously marked by both the repressive H3K9me3 and activatory H3K27ac (Figure 1c, d).

Minor issues:

- What is y axis of Fig5C? Unclear at present.

Response: y-axis is the RNA-seq expression level of the genes at the indicated embryonic stage. We have reworded the figure legend to make this clearer.

-Please show genome assembly version together with coordinates, in appropriate Figures.

Response: We have added the genome assembly to the figures.

-Figure S5 - What are the columns? What does the red line represent? Legend is not informative here.

Response: We thank for reviewers point out this, Figure S5a columns represent the percent of reads mapping to the TEs, for the RNA-seq data for each KD. The redline is the % mapping for the *shLuc* sample. We have amended the figure legend.

-Fig 1B and others: Where appropriate, it would help to display the n number representing the number of unique TE instances that each figure is displaying.

Response: We have added the n= numbers to the indicated figures, where appropriate.

-Line 323 - numerous, not numerical.

-Line 441 - text missing.

Response: We have corrected these errors.

Reviewer #2 (Remarks to the Author):

He et al have performed a large-scale data mining effort to identify chromatin signatures associated with TEs in mouse ES cells, finding a variety of profiles across different subfamilies.

Knockdown of a panel of TE-associated chromatin modifiers uncovered class-specific regulators of TE transcription.

This study is most useful in summarising the TE epigenomic landscape and in providing a useful resource in the form of ATAC-seq and RNA-seq data from knockdowns of key TE regulators. What is harder to pick out is which novel results have been uncovered, as many of the highlighted findings are already known, such as: 1) the regulation of L1s and IAPs by PRDM5-mediated arginine methylation, 2) the enhancer profile of RLTR13D6 and other elements, 3) the TE classes that are regulated by SETDB1. That the emerging patterns are complex and context-specific is also not a novel insight, as the independent evolutionary paths of different TEs, along with their age, are known to lead to these different epigenetic profiles. Nevertheless, I appreciate the strength of combining data mining with the mini knockdown screen, and could see this work becoming an important reference point for anyone interested in TE epigenetics.

Response: We would like to thank the reviewer for seeing the merit in our work. To our knowledge, the epigenetic profiles of TEs has never been presented in this detail before. Additionally, whilst these things may be known amongst researchers that are familiar with TEs, outside of TE focused researchers it is poorly appreciated. When we present this work to non-TE biologists there is surprise that TE chromatin is so complex and dynamic. There remains a view in the wider genomic field that TEs are irritating background 'noise' that make a mess of the analysis of ChIP-seq. Indeed, often the first step in ChIP-seq analysis is the removal of reads mapping to repeats, or the deletion of multimapping reads. We believe our paper can be a useful counterpoint to these views, by collecting and summarising the epigenetic regulation of TEs.

Technically, however, the study suffers from major flaws in the analysis of ChIP-seq data:

1) Non-uniquely mapped reads were included in all analyses. Whilst this was not spelled out in the methods section, it is clear from the profiles displayed throughout, which do not have the expected decreased mappability across young TEs. The default output from bowtie2 will assign reads with multiple hits of equal quality to one of those locations at random. Therefore, mapping to individual TE copies is ambiguous and no claims can be made about which copies

are enriched in which marks. This sort of mapping is only acceptable when generating class-wide profiles, such as those in Figure 1b. In contrast, profiles that display individual TE copies (e.g., heatmaps in Figure 1c and browser snapshots in Figure 1d) are inaccurate and misleading. To demonstrate the difference, I have remapped the H4R3me2 data and extracted uniquely mapped reads. In the attached file I show the profile for the L1Md_T element displayed in Figure 1d (see panel A) and the heatmap for L1Md_F3 elements from Figure 1c (see panel B). Importantly, this has a major impact on any classification that relies on the % of copies that bear a given epigenetic mark, which the authors use throughout. To use this classifier the authors have to: a) use uniquely aligned reads, and b) work out the % only relative to copies that have sufficient mappability to make the claim that a given epigenetic mark is present/absent. This will inevitably affect younger elements more than older ones. Claims that RLRETN_Mm TEs bear both activating and repressive marks at the same copies are also potentially affected by this.

Response: We thank the reviewer point out this, and agree that our original measure of ‘% elements marked’ is misleading. We have changed our analysis method, and we treat all TEs as ‘metagenes’ in the paper and generate class-wide profiles. We also reduce the emphasis in the paper on specific TE copies, and have deleted **Supplementary Figure 6**, for which the conclusions were unreliable without precise mapping of TE copies. This change in technique has resulted in us changing the measure we use to a fold-enrichment score, as used in (Day et al., 2010; Elsasser et al., 2015; Goldberg et al., 2010; Wu et al., 2016). We use a few individual genomic loci to illustrate the data, as we find these are helpful for the reader to visualize the genomic impact of the chromatin regulation of the TEs. This is in line with many other publications that deal with multimapped reads in a similar way to us (Bulut-Karslioglu et al., 2014; Theunissen et al., 2016). We have altered the methods section to make it clearer how we are dealing with multimapped reads (changes marked in red in the text), and we have added text to the results section that describes issues of mapping ambiguity in TEs (changes marked in red in the text).

For simultaneous marking of TEs by both activatory and repressive marks, we have added a new section in which we perform native-ChIP-qPCR with primer pairs that target a single genomic locus (**Figure 1c, d**). This data highlights that, at least for these examples, TEs can harbor bivalent repressive/activatory marks at the same time.

2) No peak calling algorithm was used. Even though it is missing from the methods section, the legend to Figure 1 states that TEs were classified as marked based on a cut-off of reads per million (but not normalized to TE length?). The problem with this approach is that it dampens the signal from localized peaks within TEs and does not take into account the background from an input sample. For example, the 5' UTR of L1 elements harbours many peaks for proteins associated with its regulation (see, e.g., PMID 29802231). Panel C of the attached figure shows how 5hmC (from the same dataset used by the authors) is highly enriched at full-length L1Md_T elements, yet was not picked up by the authors as an enriched mark. I also would have expected to see enrichment for at least H3K4me3 and H3K27ac at the same elements. Marking of a given TE copy should be based on peak calling, using the appropriate input samples.

The ATAC-seq data generated by the authors are an exception and did involve peak calling, but it looks again like this was based on non-uniquely mapped reads.

Response: Because ChIP-seq peak calling algorithms rely on the patterns at individual TE copies, and due to the unreliability of using individual elements (as highlighted above), we have altered the analysis to only consider TEs as 'metagenes', and to use fold-enrichment (observed/expected) versus the background as our analysis metric. We now mimic peak discovery within the TEs, by taking the maximum signal in any 500 bp bin across the TE, as our enrichment score. In this way it is similar to peak calling, as it allows us to detect localized enrichment within longer TEs (See for example **Supplementary Fig S1d, S1e**, and this extends to the analysis of all chromatin marks, chromatin modifiers and ATAC-seq data in the manuscript) This change in analysis technique can now correctly detect multiple chromatin marks at L1Md_T (**Supplementary Fig S1d, S1e**).

For RPM, we would like to apologize as we did not make it clear that all of the measures in the paper were already normalized to TE length, and should have been described as 'RPKM'. We have corrected this in the revised manuscript.

3) Full-length and truncated elements are analysed together. One cannot generalize the pattern of a TE class when full-length elements are pooled together with much shorter

elements that bear only part of the coding or regulatory regions of that TE. This affects L1 elements most prominently, as the vast majority of truncated elements lacks the regulatory 5'UTR. Therefore, 5hmC enrichment can be seen at full-length elements, but not truncated ones (see panel C of the attached figure), and the same will be true of other marks associated with the 5' UTR. This is less of an issue with LTR elements, as the Repeatmasker annotation separates LTR ends from the coding region.

Response: We are grateful for reviewer point out this. We have added a new section in the manuscript which specifically discusses chromatin at full length and truncated LINE L1 elements.

During the change of methods from a '% marked' to a 'fold-enriched', we also took the opportunity to change how we detected enrichment. We now take the maximum scoring 500 bp bin window across the metagene for each TE type. This has the effect of more accurately detecting localized enrichments within the TE, particularly for longer TEs.

The manuscript therefore needs a major reshuffle of its bioinformatic approaches, to avoid becoming a minefield of artefacts associated with the complexity of analyzing short sequencing data from TEs.

Response: We would like to thank the reviewer for their insightful comments, which we have used to improve the robustness of the bioinformatic analysis. We have tried to accommodate all of reviewer 1 and 2's comments as best we can, however there is some conflict between the advice given by each reviewer. Reviewer 1 asked us to expand the analysis of individual elements, whilst reviewer 2 cautioned us against the ambiguities of analyzing individual elements. We have tried to reconcile these differences, and our current analysis is in line with common practice in the field, and closely follows analysis techniques published in (Bulut-Karslioglu et al., 2014; Day et al., 2010; Elsasser et al., 2015; Goldberg et al., 2010; Theunissen et al., 2016; Wu et al., 2016), and from other leaders in the field.

Reviewer #3 (Remarks to the Author):

This manuscript provides data relevant to mechanisms involved in the substantial context-specificity of the chromatin modification system, which marks and regulates TE using

a regulatory chromatin code. The authors showed that TEs are marked not only by heterochromatic marks, but are labelled by all major types of chromatin modification in complex patterns in mESC. They knocked-down 41 chromatin modifiers and found 29 significantly deregulated at least one type of TE. Significant effect was found in the loss of Setdb1, Ncor2, Rnf2, Hdac5, Prmt5, Kat5, Uhrf1, Rrp8 and Ash2l, which caused widespread changes in TE expression and matching changes in chromatin opening. These effects were context-specific, as different chromatin modifiers regulated the expression and chromatin accessibility of specific subsets of TEs. Some of the content are relative novel, however, the manuscript covers many problems and needed to be largely and carefully improved.

Major points:

1. The novelty is moderate. The epigenetic regulation of transcription, and the relationship between the chromatin marks and the chromatin factors are well known and not novel at all.

Response: This comment was overruled by the editor.

2. The functional validation of TE regulation is lacked. Does the changed of TEs caused by CMs affect genome stability?

Response: This is an excellent idea from the reviewer, but we believe this is beyond the scope of the current manuscript. To detail this, we would need knockouts or prolonged CM knockdowns which would need to be followed up with detailed genome wide sequencing, karyotyping and copy number variation measurements. As our current manuscript focuses on the short-term deregulation of TEs, we prefer to leave this work for future studies. We speculate that the answer to this question would be yes, for example, based on the Setdb1^{-/-} embryos, which show genome instability. We think this would be an excellent follow up study.

3. The information in the introduction is too simple and incomplete. The information about TE function, regulatory mechanism and TE-related diseases are not included. The previous publications about the relationship between TE and epigenetic regulation are not described. The current achievement and key issues needed to be solved are not clear.

Response: We have rewritten the introduction according to the reviewer, adding in extra descriptions of TE functions, their regulatory mechanism, and known epigenetic regulation. We have focused the abstract to more clearly communicate the key issues and the results.

4. This manuscript shown that TEs are also marked by active chromatin marks, such as H3K4me1, and H3K4me3. Moreover, the expression of the TEs are down regulated when CMs are knocked down. The regulatory mechanisms of these TEs have not been studied and explained. Are they also suppressed in ES cells?

Response: We were interested to explore this, it was surprising to us that only a few TEs were down-regulated. One possible explanation for this is that our miniscreen did not include enough activators, or that activators are redundant and several KDs are required to see an effect. Another explanation is that the TEs are marked in a bivalent pattern, with both activatory and repressive marks, and manipulating one or other may not lead to repression. We have added text to the discussion (changes marked in red in the text).

Due to the lack of a large number of down-regulated TEs, we are unable to perform a global analysis. However, we have briefly explored this issue for *Chd4* (see figure below), which has the largest number of down-regulated TE types. Briefly, we see a selection of TE types that are downregulated, including some LINE and IAP elements, but mostly ERVs. CHD4 acts as a cofactor for the NuRD complex, and does not have direct catalytic activity, so linking CHD4 to specific chromatin marks is difficult. Additionally, *Chd4* KD showed strong signs of differentiation (**Supplementary Fig. S4e, S4f**), which makes any analysis of this KD challenging. Consequently, we have not added this analysis to the manuscript. We hope to explore the downregulation of TEs more fully in other studies, if we can discover chromatin regulators that lead to the widespread downregulation of TEs. Due to the lack of robust down-regulation of TEs we do not explore this in the current manuscript, beyond the comments we have added into the discussion.

Figure 1. Down-regulated TEs in the shChd4 KD. (a) Fold-change of the significantly down-regulated TE types in shChd4 KD. (b) Boxplots of example TE types that are downregulated in the Chd4 KD. (c) Example genomic views of TEs that are down-regulated in shChd4 KD.

5. As shown in the manuscript, in agreement with the prevalence of chromatin marks at TEs, CMs were also often bound to TEs in specific patterns. Is the pattern of CMs consistent with chromatin markers? The manuscript gives a few examples, however, the global analysis is lacking.

Response: The global analysis of CMs versus chromatin marks at TEs is now presented in a revised **Supplementary Figure 3**, and an expanded description (changes marked in red in the text). This figure shows that chromatin marks at TEs are often co-correlated in supergroups, and subgroups that are dominated by known TE-regulators and many novel factors. We can observe a heterochromatin group which contains SETDB1, H3K9me3 and H4K20me3, and active/open groups that contain ATAC data, H3K27ac and activators such as p300. This indicates that the pattern of CMs is consistent with the chromatin marks. We thank the reviewer for giving us the opportunity to refocus this part of the paper, which we now believe presents the global patterns of CMs and chromatin marks at TEs in a unified manner.

6. To rule out the change of cell fate in response to CM KD, the detection of pluripotency and differentiation gene expression is not enough. Protein expression level and ES cell phenotype should be detected.

Response: We have clarified this section, with an expanded discussion about *shChd4* and *shMcrs1* which do show signs of differentiation. We have added Western blots for the KDs that we did ATAC-seq for (**Supplementary Fig. S5c**), there was only small changes in the levels of the pluripotency markers OCT4 or NANOG. If there were any robust signs of differentiation NANOG and OCT4 would be completely lost within 4 days.

7. To prove the relationship between TE expression and chromatin markers, or the relationship between the change of chromatin markers and chromatin state after the depletion of CMs, there is a lack of global wide statistics. The current data are difficult to explain it adequately, because the examples given are different in two parts.

Response: We thank the reviewer for this comment. We have been careful in making claims about the relationship between TE expression and chromatin marks, as the manuscript has no ChIP-seq in the KDs. Identifying the chromatin marks that are altered in the KDs is challenging, as many of the chromatin modifiers are either redundant with other enzymes (e.g. SETDB1, G9A, SUV39H1), are part of multi-protein complexes (RNF2, SETDB1, HDACs), or are co-factors with no catalytic domains (NCOR1/2, ASH2L). Consequently, KD of a single chromatin modifier could affect a wide range of chromatin marks, particularly as compensation mechanisms are known to be activated, for example (Berrens et al., 2017), making their global analysis challenging. Whilst it would be desirable to add ChIP-seq of specific chromatin marks, in practice a large number of both direct and indirect chromatin marks would need to be performed and this would need to be done on a case-by-case basis. This makes ChIP-seq prohibitive if we aim to globally survey chromatin changes. Consequently, we decided to use ATAC-seq as a proxy for changes in chromatin marks, this allowed us to uniformly analyze chromatin changes across the different KDs.

8. In the part of that chromatin modifiers alter chromatin accessibility at specific sets of TEs, the TEs with up-regulated expression and opened chromatin are described. It is necessary to describe the TEs with down-regulated expression and closed chromatin.

Response: This comment is related to comment 4 above. Whilst we find it extremely interesting to probe down-regulated TEs, we do not have enough of them in our dataset to draw meaningful conclusions.

9. “we chose to KD over a short time period of 4 days.....” at page 8. 4 days after KD is the standard time point to check ESC differentiation. The authors should provide the reasons and evidence why they chose the time point.

Response: We have revised this part of the text to make the choice of day 4 more clear:

“As cells can compensate for the derepression of TEs by activating alternative suppressive mechanisms^{44,46}, we chose to knockdown over a short time period of 4 days. This decision has two benefits. First, it allowed us to study those CMs that are lethal when knocked down for a prolonged time. Second, it should minimize changes in cell fate which would complicate the results as the cells reconfigure their chromatin in response to differentiation”

Minor points:

1. The manuscript could be written in a more concise manner to transmit clearer its important message. The English should be largely revised, for example at page 2, potential danger to genomic stability or instability? Page 3. Cell’s DNA should be instead by cellular genome. Page 10, “although” and “nonetheless” should not coexist in one sentence. Page 11, “looking at” is so informal in manuscript.....I hope the author should pay more attention to improve the whole English grammar and the verbal accuracy, not only revise what I mentioned here.

Response: We have worked through the paper to streamline the English. We thank the reviewer for pointing out these specific errors, and we have corrected or rephrased them.

2. There are many sentences with grammar mistake, which need to be carefully revised by the author. Such as, page 2, “particularly potent” is an adjective and cannot be used as subject. Page 7, “As many of the IAPs had methylated DNA (Fig.1c), and, as previously reported, these IAPs were bound by MBD-family proteins.” In page 7, “also” appeared twice in the next sentence. The meaning of “Overall the RLTR46.....of the TE RLTR46 copies” is confused. Page 13, “of the KDs,” also has serious grammar mistake. Page 14, “Although whether all

of these pathways...". "although" cannot be used in the front of an independent sentence. The author should also pay attention to the repeated use of words. Word's diversity should be encouraged. Page 7, "as previously reported" appeared twice nearby.

Response: We thank the reviewer for pointing out these grammar mistakes, which we have corrected. We have also carefully checked the manuscript, and have had it reviewed by our colleagues for other possible problems.

3. In page 11, the author only provides examples that KD specific CM would open chromatin and up-regulate corresponding TEs. Does it mean all CMs negatively regulate TEs and maintain compact chromatin? If not, the author should also give opposite examples.

Response: As mentioned above to point 4, we were surprised by the relatively few TE types that were negatively regulated. As we do not have a large number of examples of TEs that are down-regulated it is difficult for us to comment on any mechanisms regulating them. Ultimately, we did not find any examples where the KD closed chromatin, and down-regulated TEs. This may be due to the choice of CMs that we KD, or may be an underlying property of the regulation of TEs. We will need genome-wide KD screens, and combinatorial KD of activator CMs to address this issue. We believe this is beyond the scope of the current manuscript.

4. Page 14, it is improper to say that "Chaf1a/b, Chd5, Rnf2, Rif1, Kdm1a, Ehmt2, Yy1, Rybp or miR-34a" are pathways.

Response: We have corrected this to read 'factors'

5. The author should pay attention to gene versus protein terminology.

Response: We use the MGI standard gene symbols, and have corrected genes to Capital case italics (*Nanog*), and proteins to upper case (NANOG).

6. Line 168, the protein pol II should be written as Pol II.

Response: We have corrected this.

7. Page 3 line 49: A correct punctuation mark is missed.

Response: We have corrected this.

8. Fig.1C, the authors should provide the color bar and scale bar, as well as for other heatmaps.

Response: We apologize for these omissions. Due to a reorganization of the methods, the only remaining pileup heatmap is in Supplementary Fig. 1f. We have added the missing color bar scales for Fig 1c, S2a, S2c.

9. Fig1D and 3E, the authors should provide the range value of histogram.

Response: The range values go from a baseline of 0, up to the indicated value on the right sides of the histograms. We have added this detail into the figure legend.

10. Figure legends:

10.1 For Fig.2, some of the information was not complete.

(a) The author should mark the X- and Y-axis.

(b) Epigenetic marks are pink.

5.2 For fig.S4b and S5a, "sh" should be added in the each gene name at X-axis.

5.3 For fig.3b and 4a, the legends should be coincident with the figures. The Y-axis of fig.3b, FC or log₂ (FC)? And RPM or log₂ (RPM) For the label of fig.4a?

Response: We would like to thank the reviewer for carefully reviewing our manuscript. We have corrected these mistakes. For Figure 2a, there are too many rows to present the TE names at the minimum font size of 5pt. The complete table is presented in **Supplementary Table 3**.

References

Berrens, R.V., Andrews, S., Spensberger, D., Santos, F., Dean, W., Gould, P., Sharif, J., Olova, N., Chandra, T., Koseki, H., *et al.* (2017). An endosRNA-Based Repression Mechanism

Counteracts Transposon Activation during Global DNA Demethylation in Embryonic Stem Cells. *Cell Stem Cell* 21, 694-703.e697.

Bulut-Karslioglu, A., De La Rosa-Velazquez, I.A., Ramirez, F., Barenboim, M., Onishi-Seebacher, M., Arand, J., Galan, C., Winter, G.E., Engist, B., Gerle, B., *et al.* (2014). Suv39h-dependent H3K9me3 marks intact retrotransposons and silences LINE elements in mouse embryonic stem cells. *Molecular cell* 55, 277-290.

Day, D.S., Luquette, L.J., Park, P.J., and Kharchenko, P.V. (2010). Estimating enrichment of repetitive elements from high-throughput sequence data. *Genome biology* 11, R69.

Elsasser, S.J., Noh, K.M., Diaz, N., Allis, C.D., and Banaszynski, L.A. (2015). Histone H3.3 is required for endogenous retroviral element silencing in embryonic stem cells. *Nature* 522, 240-244.

Goldberg, A.D., Banaszynski, L.A., Noh, K.M., Lewis, P.W., Elsaesser, S.J., Stadler, S., Dewell, S., Law, M., Guo, X., Li, X., *et al.* (2010). Distinct factors control histone variant H3.3 localization at specific genomic regions. *Cell* 140, 678-691.

Guo, G., and Smith, A. (2010). A genome-wide screen in EpiSCs identifies Nr5a nuclear receptors as potent inducers of ground state pluripotency. *Development* 137, 3185-3192.

Medvedeva, Y.A., Lennartsson, A., Ehsani, R., Kulakovskiy, I.V., Vorontsov, I.E., Panahandeh, P., Khimulya, G., Kasukawa, T., Consortium, F., and Drablos, F. (2015). EpiFactors: a comprehensive database of human epigenetic factors and complexes. *Database : the journal of biological databases and curation* 2015, bav067.

Theunissen, Thorold W., Friedli, M., He, Y., Planet, E., O'Neil, Ryan C., Markoulaki, S., Pontis, J., Wang, H., Iouranova, A., Imbeault, M., *et al.* (2016). Molecular Criteria for Defining the Naive Human Pluripotent State. *Cell Stem Cell* 19, 502-515.

Waterston, R.H., Lindblad-Toh, K., Birney, E., Rogers, J., Abril, J.F., Agarwal, P., Agarwala, R., Ainscough, R., Alexandersson, M., An, P., *et al.* (2002). Initial sequencing and comparative analysis of the mouse genome. *Nature* 420, 520-562.

Wu, J., Huang, B., Chen, H., Yin, Q., Liu, Y., Xiang, Y., Zhang, B., Liu, B., Wang, Q., Xia, W., *et al.* (2016). The landscape of accessible chromatin in mammalian preimplantation embryos.

Nature 534, 652.

Reviewer #1 (Remarks to the Author):

The Authors have settled all of my concerns, and I now consider the manuscript ready for publication.

Reviewer #2 (Remarks to the Author):

The authors have done a great (and very extensive!) job re-analysing the data to take into account the caveats of read mapping and of uneven distribution of reads across TEs. I can now take the authors' conclusions with much more confidence.

I still have an issue with displaying genome browser snapshots with ambiguously mapped data (even if others have done the same in other publications), but at least the authors have highlighted this caveat in the main text. Perhaps it would be useful to re-iterate it in all the figure legends where this applies.

Reviewer #3 (Remarks to the Author):

I commend the authors on their efforts to reply to the suggestions regarding the structure of the manuscript and additional data requested. The clarity of the manuscript has greatly improved. Description of the ChIP-seq data and gene expression is more detailed and improved. This study provides relevant information regarding that the chromatin modifying enzymes manage the expression of TEs, by context-specific deposition of chromatin marks that regulate the chromatin state and expression level of the TEs. However, to further improve the manuscript, some problems need to be carefully solved.

1. Since the genome stability is the important biological function of the transposable elements, to check the effect of CMs on the genome stability through TEs will improve the significance of the manuscript. The authors do not need to do a very deep study to know the mechanism, but at least provide some evidence to show the basic effect of the mechanism the authors found on genome stability.

2. As shown in the manuscript, the role of CMs in TE expression is that CMs regulate the expression of TEs in a type-specific manner. However, the roles of chromatin markers in TE expression are not been identified. In other words, why are these chromatin modifications needed to deposit on TE?

3. The authors have revealed many relationships between the depletion of CMs and the upregulation of MERVL, which corresponds to the increased expression of 2C-stage embryonic genes. Does the depletion of CMs induce the expression of MERVL, and increase the expression of 2C-stage embryonic genes?

-

P.S. To help the scientific community achieve unambiguous attribution of all scholarly contributions, Nature Communications encourages all authors to create and link an ORCID identifier to their account. Please ensure that all co-authors are aware that they can add their ORCIDs to their accounts, so that it will display on this paper. If they so wish, they must do so before the paper is formally accepted. It will not be possible to add ORCIDs post-acceptance, e.g. at proof. To add an ORCID please follow these instructions:

1. From the home page of the <http://mts-ncomms.nature.com/cgi-bin/main.plex> MTS click on 'Modify my Springer Nature account' under 'General tasks'.

2. In the 'Personal profile' tab, click on 'ORCID Create/link an Open Researcher Contributor ID (ORCID)'. This will re-direct you to the ORCID website.

3a. If you already have an ORCID account, enter your ORCID email and password and click on 'Authorize' to link your ORCID with your account on the MTS.

3b. If you don't yet have an ORCID account, you can easily create one by providing the required information and then clicking on 'Authorize'. This will link your newly created ORCID with your account on the MTS.

Reviewers' comments:

Reviewer #1 (Remarks to the Author):

The Authors have settled all of my concerns, and I now consider the manuscript ready for publication.

Reply: We thank the reviewer for their time and effort in assessing our manuscript.

Reviewer #2 (Remarks to the Author):

The authors have done a great (and very extensive!) job re-analysing the data to take into account the caveats of read mapping and of uneven distribution of reads across TEs. I can now take the authors' conclusions with much more confidence.

I still have an issue with displaying genome browser snapshots with ambiguously mapped data (even if others have done the same in other publications), but at least the authors have highlighted this caveat in the main text. Perhaps would be useful to re-iterate it in all the figure legends where this applies.

Reply: We thank the reviewer for their positive comments on our manuscript. We have followed the reviewer's suggestion and have added a statement to the figure legend of figure 2d and Supplementary Figure 1g (the first time that ambiguous genomic views appear; note that the views in Figure 1c were validated by the native-ChIP-qPCR). However, due to space requirements in the figure legends, we do not mention this in all the figure legends.

Reviewer #3 (Remarks to the Author):

I commend the authors on their efforts to reply to the suggestions regarding the structure of the manuscript and additional data requested. The clarity of the manuscript has greatly improved. Description of the ChIP-seq data and gene expression is more detailed and improved. This study provides relevant information regarding that the chromatin modifying enzymes manages the expression of TEs, by context-specific deposition of chromatin marks that regulate the chromatin state and expression level of the TEs. However, to further improve the manuscript, some problems need to be

carefully solved.

Reply: We thank the reviewer for their positive comments on our manuscript, and their careful review that has enabled us to improve our manuscript.

1. Since the genome stability is the important biological function of the transposable elements, to check the effect of CMs on the genome stability through TEs will improve the significance of the manuscript. The authors do not need to do very deep study to know the mechanism, but at least provide some evidence to show the basic effect of the mechanism the authors found on genome stability.

Reply: We understand the reviewers interest in this, however we have not performed the correct experiments to address this thoroughly. We tried to look in our existing RNA-seq and ATAC-seq for single nucleotide variants, and known polymorphisms in that could be used to estimate copy number expansions, and so estimate genome stability. However, we do not have confidence in this analysis and we do not wish to comment on this issue in the current manuscript. We have many ideas to address this issue in future work, for example by sequencing TE integrants in a high throughput manner to track TE expansion in epigenetically impaired cells. Or by performing retrotransposition assays for chromatin modifier-specific TEs. However, this work is at a very primitive stage, and we are currently dealing with many technical challenges.

2. As shown in the manuscript, the role of CMs in TE expression is that CMs regulate the expression of TEs in a type-specific manner. However, the roles of chromatin markers in TE expression are not been identified. In other words, why are these chromatin modifications needed to deposit on TE?

Reply: This is an fascinating question that goes far beyond the scope of our manuscript. Exactly *why* TEs need to be marked by such a rich pattern is entirely unclear. There is no instinctive reason the cell should go to all the trouble of having such an elaborate system to mark chromatin at ostensibly non-functional TEs. The naïve response of the cell should be to non-discriminately silence all TEs uniformly, and so avoid all danger to genome integrity posed by TEs. However, this does not happen, instead quite the

reverse and there as a complex, and even somewhat wasteful, regulation of TEs. Some of these observations can be explained by the fact TEs are not passive bystanders, but are in active competition with the host cell (Schlesinger and Goff, 2015). Similarly, it is supposed that the TEs have beneficial roles, and can act as an evolutionary substrate (Bourque et al., 2008; Shapiro, 2017). In the plant field, where the study of TEs is often richer than that in mammalian cells, it has been suggested that the entire purpose of the epigenetic system is to manage TEs, and other functions in development have been co-opted by the cell (Lisch and Bennetzen, 2011). Ultimately, this is a fascinating question that we are not yet capable of giving a definitive answer to.

3. The authors have revealed many relationships between the depletion of CMs and the upregulation of MERVL, which corresponds to the increased expression of 2C-stage embryonic genes. Does the depletion of CMs induce the expression of MERVL, and increase the expression of 2C-stage embryonic genes?

Reply: Yes, although as many other studies have described, the relationship is not absolute, and 2C genes can be upregulated in the absence of MERVL expression and vice versa. Ultimately, the precise link between the loss of CMs, and upregulation of MERVLs and 2C stage embryonic gene remains unclear. This could be addressed with careful time course knockdowns, and single cell tracing data for the CMs that we and others have identified as regulating MERVLs and 2C genes. However, we have not performed these experiments, and as the 2C is not the main focus of the manuscript we do not expand upon it. This question was partially addressed in the recent publication from the Torres-Padilla group (Rodriguez-Terrones et al., 2018), their work suggests that *Zscan4* (a key 2C gene) precedes MERVL expression, but is not a determinant for the 2C-like cell state. Despite the flurry of papers exploring these enigmatic cells, there remains considerable work to establish the mechanisms by which they are controlled.

References

Bourque, G., Leong, B., Vega, V.B., Chen, X., Lee, Y.L., Srinivasan, K.G., Chew, J.L., Ruan, Y., Wei, C.L., Ng, H.H., *et al.* (2008). Evolution of the mammalian transcription factor binding

repertoire via transposable elements. *Genome research* 18, 1752-1762.

Lisch, D., and Bennetzen, J.L. (2011). Transposable element origins of epigenetic gene regulation. *Current opinion in plant biology* 14, 156-161.

Rodriguez-Terrones, D., Gaume, X., Ishiuchi, T., Weiss, A., Kopp, A., Kruse, K., Penning, A., Vaquerizas, J.M., Brino, L., and Torres-Padilla, M.E. (2018). A molecular roadmap for the emergence of early-embryonic-like cells in culture. *Nat Genet* 50, 106-119.

Schlesinger, S., and Goff, S.P. (2015). Retroviral transcriptional regulation and embryonic stem cells: war and peace. *Molecular and cellular biology* 35, 770-777.

Shapiro, J.A. (2017). Exploring the read-write genome: mobile DNA and mammalian adaptation. *Critical Reviews in Biochemistry and Molecular Biology* 52, 1-17.